# Geometric Algebra-Enhanced Bayesian Flow Network for RNA Inverse Design

**Rubo Wang[1, 2], Xingyu Gao[1, 2,][*] Peilin Zhao[3]**

[1]Institute of Microelectronics, Chinese Academy of Sciences, Beijing, China
[2]University of Chinese Academy of Sciences, Beijing, China
[3]School of Artificial Intelligence, Shanghai Jiao Tong University, Shanghai, China
wangrubo@hotmail.com, gxy9910@gmail.com, peilinzhao@sjtu.edu.cn

## Abstract

With the development of biotechnology, RNA therapies have shown great potential. However, different from proteins, the sequences corresponding to a single RNA three-dimensional structure are more abundant. Most of the existing RNA design methods merely take into account the secondary structure of RNA, or are only capable of generating a limited number of candidate sequences. To address these limitations, we propose a geometric-algebra-enhanced **B**ayesian **F**low **N**etwork for the inverse design of **R**NA, called **RBFN**. RBFN uses a Bayesian Flow Network to model the distribution of nucleotide sequences in RNA, enabling the generation of more reasonable RNA sequences. Meanwhile, considering the more flexible characteristics of RNA conformations, we utilize geometric algebra to enhance the modeling ability of the RNA three-dimensional structure, facilitating a better understanding of RNA structural properties. In addition, due to the scarcity of RNA structures and the limitation that there are only four types of nucleic acids, we propose a new time-step distribution sampling to address the scarcity of RNA structure data and the relatively small number of nucleic acid types. Evaluation on the single-state fixed-backbone re-design benchmark and multi-state fixed-backbone benchmark indicates that RBFN can outperform existing RNA design methods in various RNA design tasks, enabling effective RNA sequence design.

## 1 Introduction

Ribonucleic acid (RNA), as one of the fundamental biomolecule in life activities, not only undertakes the function of genetic information transmission (such as mRNA, tRNA, and rRNA), but also participates in complex biological processes such as gene expression regulation, protein synthesis, and cell signal transduction through non-coding RNAs (such as miRNA, siRNA) [1, 2, 3, 4, 5]. The realization of its functions highly depends on the secondary structures (such as stem-loops, pseudoknots) formed by its dynamic folding and the tertiary spatial conformation. RNA can be used for drug target screening [6], vaccine development (such as the COVID-19 mRNA vaccine [7]), and synthetic biology circuit design [8]. In addition, the application of RNA design in disease treatment is becoming increasingly widespread. For example, pathogenic genes can be silenced through siRNA [9], metabolic pathways [10] can be regulated by designing riboswitches [11], and highly sensitive biosensing can be achieved by using aptamers.

Despite RNA's critical biological significance, structural modeling research remains predominantly protein-centric, as evidenced by recent advances in geometric deep learning [12, 13]. This disparity arises from two key factors: there are a large number of protein three-dimensional structures in the

---

[*]Corresponding author.

PDB, and the development of some breakthrough protein algorithms [14, 15] has further promoted the development of the protein field, especially the emergence of AlphaFold2 [16]. However, for RNA, there are not many three-dimensional structures. Although some RNA structure prediction methods have emerged currently, due to the limitation of data, they cannot achieve a very high accuracy, which makes RNA design more difficult, and deep learning has not been fully utilized in RNA design. Most of the existing RNA design tools only focus on the secondary structure of RNA [17], and for the three-dimensional structure, most of them need to manually construct features [18, 19], making the design inefficient and complex. In addition, since the functions of RNA are more complex compared to proteins, the conformation of RNA is relatively flexible, and the same RNA may correspond to different conformations [20, 21, 22]. This makes it necessary to consider more factors when designing RNA compared to designing proteins, and it is difficult to directly transfer relevant algorithms. At present, certain approaches leveraging graph neural networks to model RNA for design have emerged [23, 24]. However, the majority of these methods are adapted from the protein domain and do not adequately account for the unique characteristics of RNA structures and sequences. Specifically, for a three-dimensional structure of RNA, there exists a more diverse set of corresponding sequences [19], which leads to certain limitations in practical applications.

To overcome these limitations, this paper presents **RBFN**, a geometric-algebra-enhanced Bayesian Flow Network designed for RNA inverse design. RBFN leverages geometric algebra to improve the modeling of RNA three-dimensional structure information. Simultaneously, it employs the Bayesian Flow Network to sample sequences within the parameter space (distribution). It can generate corresponding nucleotide sequences given one or more 3D backbone structures. The main contributions of RBFN include the following aspects:

- *Modeling structures using geometric algebra.* To the best of our knowledge, RBFN is the first approach to utilize geometric algebra to enhance RNA structure modeling. Using geometric algebra can effectively incorporate structural information into the scalar features of nucleotides, thus achieving more effective modeling capabilities.

- *Generation using Bayesian Flow Network.* RBFN uses the Bayesian Flow Network to generate nucleotide sequences, directly learning the distribution of nucleotides at each position and achieving sequence generation by aligning distributions, rather than operating on discrete sequences.

- *Proposing a new time-step sampling distribution.* In view of the fact that there are only four types of nucleotide sequences, a new time-step distribution is proposed, enabling the network to pay more attention to the samples of the sender distribution, so as to enhance the network's global generation ability.

- *Higher performance in RNA design.* We compare RBFN with the state-of-the-art deep learning method gRNAde and the state-of-the-art physics-based tool Rosetta. The experimental results show that the sequences generated by RBFN have a higher sequence recovery rate, indicating that RBFN can better learn the relationship between structures and sequences.

## 2 Background

In this section, we introduce the definition of the problem in Section 2.1, introduce geometric algebra in Section 2.2, and introduce Bayesian Flow Network [25] in Section 2.3.

### 2.1 Problem Definition

RNA inverse design can be formulated as a conditional generation task: design the corresponding RNA sequence under the condition of a given backbone structure. Figure 1 shows the backbone structure of RNA. RNA is composed of four nucleotides, namely adenine (A), cytosine (C), guanine (G), or uracil (U). We follow the representation method of existing work [24] and represent RNA in the form of a Graph. In the backbone atoms of RNA, P, C4', N1 (pyrimidine) or N9 (purine) are

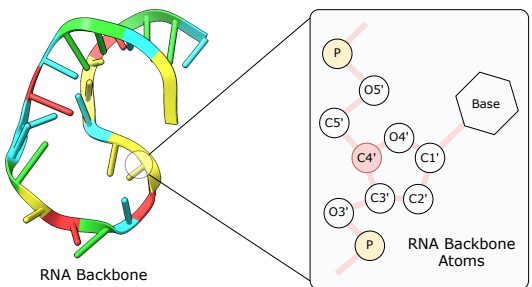

Figure 1: Schematic diagram of RNA backbone structure. RNA is composed of four basic nucleotides. The backbone atoms of each nucleotide can be represented by the structural form on the right.

retained as backbone atoms. With this representation, the RNA backbone can generally be fully represented, and at the same time, it can prevent representation problems caused by excessive torsional space. Each nucleotide $i$ serves as a node, and the three-dimensional coordinate $\vec{x}_i \in \mathbb{R}^3$ is the center of the backbone atoms of the entire nucleotide. When given a three-dimensional backbone structure of RNA, the goal is to design a **nucleotide sequence** that can fold into this three-dimensional structure.

## 2.2 Geometric Algebra

Geometric algebra refers to the Clifford algebra over a real vector space. It can describe objects such as points, lines, and planes and operations on them in an algebraic form, and is a very powerful mathematical framework. Algebraic elements are divided into homogeneous components according to their orders: a $k$-vector is a linear combination of $k$-blades (basis elements of grade $k$). For example, a 1-vector (i.e., the conventional vector) is composed of 1-blades, and a bivector is composed of a linear superposition of 2-blades. More generally, a multivector $\mathbf{x} \in \mathbb{G}_{p,q,r}$ is the direct sum of subspaces of all grades, $\mathbb{G}_{p,q,r}$ denotes a geometric algebra over a real vector space with $p$ basis vectors of positive signature ($e_i^2 = +1$), $q$ of negative signature ($e_j^2 = -1$), and $r$ of zero signature ($e_k^2 = 0$), satisfying $p + q + r = n$ for an $n$-dimensional space. And can be expressed as $\mathbf{x} = \sum_{k=0}^{n}[\mathbf{x}]_k$, where $[\mathbf{x}]_k$ represents the homogeneous component of grade $k$. Taking the three-dimensional geometric algebra $\mathbb{G}_{3,0,0}$ as an example, its 8 basis blades correspond to a complete component decomposition:

$$\mathbf{x} = \underbrace{x_0 \cdot 1}_{\text{Scalar}} + \underbrace{x_1 e_1 + x_2 e_2 + x_3 e_3}_{\text{Vector}} + \underbrace{x_{12} e_{12} + x_{13} e_{13} + x_{23} e_{23}}_{\text{Bivector}} + \underbrace{x_{123} e_{123}}_{\text{Trivector}} \tag{1}$$

where $e_{ij} = e_i \wedge e_j$ ($i < j$) and $e_{123} = e_1 \wedge e_2 \wedge e_3$. The metric signature $(p, q, r) = (3, 0, 0)$ indicates three basis vectors with $e_i^2 = +1$, none with $e_j^2 = -1$, and no null vectors ($e_k^2 = 0$). Anticommutation ($e_i e_j = -e_j e_i$ for $i \neq j$) ensures closure under the geometric product, as products of basis blades reduce to linear combinations of basis blades. By choosing different parameters $(p, q, r)$, geometric algebra can efficiently model various geometric spaces such as Euclidean spaces. Its hierarchical structure provides a unified mathematical framework for the algebraic operations of geometric objects. We provide a more detailed introduction to geometric algebra in Appendix B.1.

## 2.3 Bayesian Flow Network

Bayesian Flow Networks (BFNs) [25] provide an innovative approach for generative modeling of discrete data. When dealing with discrete data, BFNs model the evolution of probability distributions through a continuous-time transport process, while maintaining mathematical rigor and computational efficiency. For discrete data with $K$ categories, BFNs start from a simple input distribution $q_0(\boldsymbol{x})$, which is chosen as a categorical distribution:

$$q_0(\boldsymbol{x}) = \text{Categorical}(\boldsymbol{x}|\pi_0) \tag{2}$$

where $\pi_0 \in \mathbb{R}^K$ is a probability vector satisfying $\sum_{k=1}^{K} \pi_{0k} = 1$ and $\pi_{0k} > 0$. BFNs introduce a continuous-time parameter $t \in [0, 1]$ and define two key distributions: (1) the *sender distribution* $q_t(\boldsymbol{x}_t)$ representing the data distribution at time $t$, and (2) the *receiver distribution* $q_t(\boldsymbol{x}_t|\boldsymbol{x}_0)$ describing the forward process from the original data $\boldsymbol{x}_0$ to the corrupted data $\boldsymbol{x}_t$. The core of BFNs is the Bayesian inversion mechanism, where the reverse process is given by:

$$q_t(\boldsymbol{x}_0|\boldsymbol{x}_t) = \frac{q_t(\boldsymbol{x}_t|\boldsymbol{x}_0)q_0(\boldsymbol{x}_0)}{q_t(\boldsymbol{x}_t)}, \quad q_t(\boldsymbol{x}_t) = \sum_{\boldsymbol{x}_0} q_t(\boldsymbol{x}_t|\boldsymbol{x}_0)q_0(\boldsymbol{x}_0) \tag{3}$$

To make this process learnable, BFNs use a neural network to approximate the reverse process $q_t(\boldsymbol{x}_0|\boldsymbol{x}_t)$. Specifically, the network outputs logits $\boldsymbol{z} = \text{NN}_\theta(\boldsymbol{x}_t, t)$, which are converted to probabilities via:

$$\hat{q}_t(\boldsymbol{x}_0|\boldsymbol{x}_t) = \text{Categorical}(\boldsymbol{x}_0|\text{softmax}(\boldsymbol{z})) \tag{4}$$

The training objective minimizes the KL divergence between true and predicted reverse processes:

$$\mathcal{L}(\theta) = \mathbb{E}_{t \sim \mathcal{U}[0,1], \boldsymbol{x}_0 \sim q_0, \boldsymbol{x}_t \sim q_t(\cdot|\boldsymbol{x}_0)} \left[ \text{KL} \left( q_t(\cdot|\boldsymbol{x}_t) \| \hat{q}_t(\cdot|\boldsymbol{x}_t) \right) \right] \tag{5}$$

For discrete data, BFNs do not use SDEs but instead employ a deterministic transport process. The marginal distribution evolves as:

$$\pi_t = (1 - t)\pi_0 + t\bar{\pi}_0 \tag{6}$$

where $\bar{\pi}_0$ is the empirical data distribution. We provide a more detailed introduction to Bayesian Flow Network in Appendix B.2.

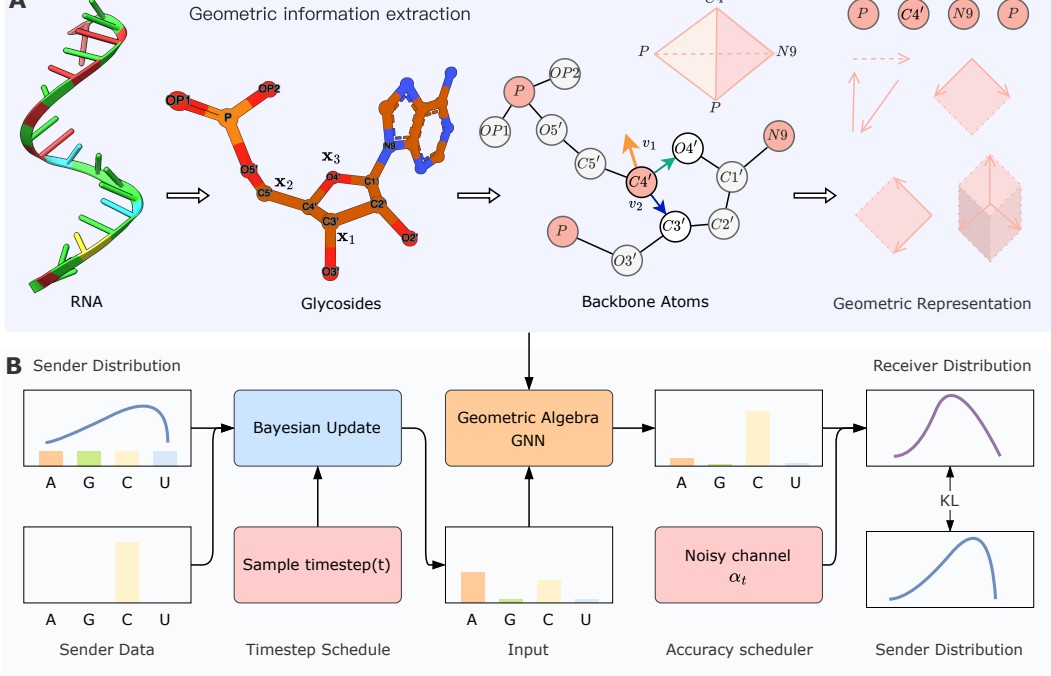

Figure 2: **The RBFN pipeline for 3D RNA inverse design. A.** The 3D structure of RNA can be split into the splicing of individual nucleotide structures. For each individual nucleotide, using geometric algebra, it can simultaneously be interpreted as group elements (0-vector, 1-vector, 2-vector, 3-vector). The structure of nucleotides is derived from [26]. **B.** The framework of RBFN. RBFN uses Geometric Algebra GNN as the backbone network, and learns the sequence distribution alignment under the RNA backbone structure through the Bayesian Flow Network. The entire update process uses KL to align the sender distribution and the receiver distribution.

## 3 Methods

We propose a geometric algebra-enhanced Bayesian Flow Network for learning the sequence distribution under the condition of the structure, enabling sequence design given the three-dimensional structure of the RNA backbone. The proposed method is an extension of the Bayesian Flow Network. We use a geometric algebra-enhanced graph neural network to enhance the modeling of RNA three-dimensional structures. At the same time, based on the characteristics of the Bayesian Flow Network, we directly learn the sequences distribution correspondence for structures. In addition, based on the characteristics of RNA sequences, we propose a time-step sampling method to better train the Bayesian Flow Network.

### 3.1 Geometric Algebra GNN for RNA Three-Dimensional Structure Modeling

As shown in the top of Figure 2, based on the three-dimensional structure of RNA, we isolate each nucleotide as a structural unit. For nucleotide $i$, the phosphate atom ($P_i$), C4' atom of ribose, glycosidic bond atom (N1 for pyrimidines or N9 for purines), and the 5' adjacent phosphate ($P_{i-1}$) form a tetrahedral geometry. Using these four atomic coordinates $\{\vec{x}_i^{(P)}, \vec{x}_i^{(C4')}, \vec{x}_i^{(N1/N9)}, \vec{x}_{i-1}^{(P)}\}$, we construct a multivector $M_i \in \mathbb{G}_{3,0,0}$, where: (1) the 0-vector encodes $P_i$-C4' distance; (2) 1-vectors represent bond directions (e.g., $\vec{x}_i^{(P)} - \vec{x}_i^{(C4')}$); (3) 2-vectors capture oriented planes (e.g., $P_{i-1}$-$P_i$-C4' plane); (4) the 3-vector represents the tetrahedron's oriented volume. Next, we construct a nucleotide graph where node $i$ is positioned at $\vec{x}_i^{(P)} \in \mathbb{R}^3$, and edges connect to the 32 nearest neighbors (selected to cover 1 helical turn in A-form RNA, 11Å radius). We expand on [24, 27] by integrating geometric algebra operations into GNN message passing: scalar features in $S$ store 0-vector components, while vector neurons process 1/2/3-vector components for SE(3)-equivariant updates. Specifically, the architecture comprises an encoder that processes multivector features through $L$ geometric algebra GNN layers, and a decoder that predicts nucleotide identities from the learned representations. Due to

RNA's conformational diversity, we implement a multi-state encoder following [24], which processes ensembles of backbone conformations to capture structural variability.

For an RNA structure with $m$ conformations and $n$ nodes, consider this structure as multiple graphs, i.e., $\{\mathcal{G}^{(1)}, \ldots, \mathcal{G}^{(m)}\}$. For each conformation graph $\mathcal{G}^{(c)} = (\boldsymbol{A}^{(c)}, \boldsymbol{S}^{(c)}, \vec{\boldsymbol{X}}^{(c)})$ where $c \in \{1, \ldots, m\}$, it contains scalar features, namely the index and sequence representation of nucleotides in RNA $\boldsymbol{S} \in \mathbb{R}^{n \times m \times f}$, and vector features $\vec{\boldsymbol{X}} \in \mathbb{R}^{n \times m \times f' \times 3}$: (a) the forward and backward unit vectors along the backbone from the 5' end to the 3' end, $(\vec{\boldsymbol{x}}_{i+1} - \vec{\boldsymbol{x}}_i$ and $\vec{\boldsymbol{x}}_i - \vec{\boldsymbol{x}}_{i-1})$; and (b) the unit vectors, distances, angles, and torsions from each C4' to the corresponding P and N1/N9. The edge features $\{\boldsymbol{A}^{(1)}, \ldots, \boldsymbol{A}^{(m)}\}$ from node $j$ to $i$ are initialized as follows: (a) the unit vector from the source node to the target node, $\vec{\boldsymbol{x}}_j - \vec{\boldsymbol{x}}_i$; (b) the distance in three-dimensional space, $\|\vec{\boldsymbol{x}}_j - \vec{\boldsymbol{x}}_i\|_2$, encoded by 32 radial basis functions; and (c) the distance along the backbone, $j - i$, encoded by 32 sine-position encodings, and finally represented as $\vec{\boldsymbol{V}}^{(c)} \in \mathbb{R}^{n \times n \times f' \times 3}$. The geometric algebra feature $\mathsf{q}(\mathbf{v})$ is obtained based on the three-dimensional coordinate relationships. For a multivector $\mathbf{v} = s + \vec{v} + \mathbf{b} + t\mathbf{I}$ where $s$ is scalar, $\vec{v}$ vector, $\mathbf{b}$ bivector, and $t\mathbf{I}$ pseudoscalar components, the quadratic form is $\mathsf{q}(\mathbf{v}) = s^2 + \|\vec{v}\|^2 - \|\mathbf{b}\|^2 - t^2$ in $\mathbb{G}_{3,0,0}$. Following [28, 27], an MVP-style message passing layer is defined as:

$$\mathbf{v}_{ij}^l = \boldsymbol{\phi}_e\left([\mathbf{v}_i^l, \mathbf{v}_j^l]\right) \tag{7}$$

$$s_{ij}^l = \phi_e\left([s_i^l, s_j^l]\right) \tag{8}$$

$$(s_{ij}^{l'}, \mathbf{v}_{ij}^{l'}) = (\text{MVP-GP}_e \circ \text{MVP-Lin}_e)\left(s_{ij}^l, \mathbf{v}_{ij}^l\right) \tag{9}$$

$$(s_i^{l'}, \mathbf{v}_i^{l'}) = \frac{1}{\sqrt{|\mathcal{N}_i|}} \sum_{j \in \mathcal{N}_i} s_{ij}^{l'}, \frac{1}{\sqrt{|\mathcal{N}_i|}} \sum_{j \in \mathcal{N}_i} \mathbf{v}_{ij}^{l'} \tag{10}$$

$$(s_i^{l+1}, \mathbf{v}_i^{l+1}) = (\text{MVP-GP}_v \circ \text{MVP-Lin}_v)\left([s_i^l, s_i^{l'}], [\mathbf{v}_i^l, \mathbf{v}_i^{l'}]\right) \tag{11}$$

Details can be found at Appendix B.1.

## 3.2 Bayesian Flow Networks for RNA Generation

The overall architecture of RBFN is shown in Figure 2. Before training, we define a **uniform prior** over the 4 RNA nucleotides: $p_0(\boldsymbol{y}) = \text{Categorical}(\boldsymbol{y}|\pi_0)$ with $\pi_0 = [1/4, 1/4, 1/4, 1/4]^\top$. During training, for each RNA backbone graph $\mathcal{G}^{(c)} = (\boldsymbol{A}^{(c)}, \boldsymbol{S}^{(c)}, \vec{\boldsymbol{V}}^{(c)})$ with $n$ nucleotides, the forward process corrupts the ground-truth sequence $\boldsymbol{y}^* \in \{1, 2, 3, 4\}^n$ to a noisy version $\boldsymbol{y}_t$ at time $t \sim \mathcal{U}[0, 1]$ via:

$$q_t(\boldsymbol{y}_t|\boldsymbol{y}^*) = \prod_{i=1}^n \left[(1 - \alpha_t)\delta_{\boldsymbol{y}_{ti}, \boldsymbol{y}_i^*} + \alpha_t \pi_{0, \boldsymbol{y}_{ti}}\right] \tag{12}$$

where $\alpha_t = t$ is the noise level, and $\delta$ is the Kronecker delta. The *sender distribution* is modeled by a geometric algebra GNN that takes as input: (1) the noisy sequence representation $\boldsymbol{S}_t \in \mathbb{R}^{n \times 4}$ (one-hot of $\boldsymbol{y}_t$), and (2) structural features from $\mathcal{G}^{(c)}$. The network outputs logits $\boldsymbol{z} = \text{GNN}_\theta(\boldsymbol{S}_t, \mathcal{G}^{(c)}, t)$, which parameterize the *receiver distribution*:

$$p_\theta(\boldsymbol{y}^*|\boldsymbol{y}_t, \mathcal{G}^{(c)}) = \prod_{i=1}^n \text{Categorical}\left(\boldsymbol{y}_i^* \mid \text{softmax}(\boldsymbol{z}_i)\right) \tag{13}$$

The training objective minimizes the KL divergence between true and predicted reverse processes:

$$\mathcal{L}(\theta) = \mathbb{E}_{t, \boldsymbol{y}^*, \boldsymbol{y}_t}\left[\sum_{i=1}^n \text{KL}\left(q_t(\cdot|\boldsymbol{y}_i^*) \,\|\, \text{softmax}(\boldsymbol{z}_i)\right)\right] \tag{14}$$

At inference, starting from $\boldsymbol{y}_1 \sim p_0$, we iteratively sample $\boldsymbol{y}_t$ from $p_\theta(\boldsymbol{y}^*|\boldsymbol{y}_{t+\Delta t}, \mathcal{G}^{(c)})$ to generate the final sequence $\hat{\boldsymbol{y}} = \boldsymbol{y}_0$. The geometric algebra features (0-3-vectors from tetrahedral geometries) are integrated into the GNN's vector neuron layers to preserve SE(3)-equivariance during message passing.

The Bayesian update function can be expressed as $\boldsymbol{S}\left(\Psi_{i-1}, \mathbf{y}, \alpha\right) = \frac{e^{\mathbf{y}}\Psi_{i-1}}{\sum_{k=1}^K e^{\mathbf{y}_k}(\Psi_{i-1})_k}$

$$p_F(\Psi^{\boldsymbol{S}_t} \mid \mathbf{h}_t; t) = \mathop{\mathbb{E}}_{\mathcal{N}\left(\mathbf{y}^{\boldsymbol{S_t}}|\beta(t)\left(K\mathbf{e}_{\mathbf{h}_t}-\mathbf{1}\right), \beta(t)K\mathbf{I}\right)} \delta(\Psi^{\boldsymbol{S}_t} - \text{softmax}(\mathbf{y}^{\boldsymbol{S_t}})) \tag{15}$$

With the help of the geometric algebra GNN $\Psi$, the output distribution can be obtained through the following expressions:

$$p_O^{(d)}(k \mid \Psi^g; t) = \left(\text{softmax}\left(\Phi^{(d)}(\Psi^g, t)\right)\right)_k, p_O(\boldsymbol{S}_t \mid \Psi; t) = \prod_{d=1}^{D} p_O^{(d)}\left(\boldsymbol{S}_t^{(d)} \mid \Psi^g; t\right) \quad (16)$$

Based on the formula derivation of [25], the unified continuous-time loss is obtained as:

$$L^\infty(\boldsymbol{S}_t \mid \vec{\boldsymbol{V}}) = K\beta(1) \underset{t \sim U(0,1), p_F(\Psi \mid \boldsymbol{S}, t)}{E} t\|\mathbf{e}_{\boldsymbol{S}} - \mathbf{e}(\Psi, t)\|^2, \quad (17)$$

where

$$\mathbf{e}(\Psi, t) \doteq \left(\mathbf{e}^{(1)}(, t), \dots, \mathbf{e}^{(D)}(\Psi, t)\right). \quad (18)$$

### 3.3 Time Sampling Distribution

Similar to diffusion models and flow matching, the Bayesian Flow Network needs to sample time steps as the basis for Bayesian updates. For continuous time, time can be sampled arbitrarily in the interval (0-1). Previous works [25, 29, 30] directly used a uniform distribution to sample from 0 to 1. In images, enhanced sampling of $t$ closer to $t = 1$ prompts the model to focus its capabilities on synthesizing accurate local details, which are generated at the end of the generation process. More sampling at smaller $t$ values can improve large-scale features. In image generation, sampling is usually increased at the intermediate $t$ [31, 32]. However, for RNA inverse design, since there are only four nucleotides (A, C, G, U), slightly adding sender noise generally has little impact on the sampled sequence. To make the training process pay more attention to the transition from the initial distribution to the true distribution, we designed a new $t$ sampling function that pays more attention to smaller $t$ values, that is, it is more inclined to the sender distribution during the Bayesian update process. The specific formula is as follows:

$$p(t) = 0.02\,\mathcal{U}(0, 1) + 0.98\,\mathcal{B}(1.0, 1.9),$$

where $\mathcal{B}(\cdot, \cdot)$ is the beta distribution, to promote the generation of accurate local details. We incorporate uniform sampling to avoid the sampling density being zero when $t \to 1$. The visualization is shown in Figure 3.

## 4 Experiments

### 4.1 Experimental Setup

**Dataset.** We use RNASolo [33] for training, validation, and testing. It contains all RNA-containing structures obtained from the Protein Data Bank (PDB). Following the partitioning in [24], we use RNA structures with a resolution of $\leq 4.0$ Å. In total, 12,011 structures are available (RNASolo data cutoff: 31 October 2023). We perform structure clustering using US-align [34] with a similarity threshold of TM-score > 0.45, and sequence clustering for RNA sequence homology using CD-HIT [35]. A total of $\sim$4000 samples are obtained and divided according to different tasks. For the Single-state split, we use the partitioning method in [36] to identify the structural clusters (including riboswitches, aptamers, and ribozymes) belonging to the RNAs identified in [36]. A total of 100 sequences are obtained for testing. Based on the remaining sequences, 100 are randomly selected as the validation set, and the rest are used as the training set. This benchmark is called the **Das Benchmark**. For the Multi-state split, we calculate the pairwise C4' RMSD of the structures corresponding to each sequence. The top 100 samples from clusters with the highest median intra-sequence RMSD are added to the test set, which is called the **Multi-state Benchmark**. The next 100 samples are added to the validation set, and the rest are used as the training set for training.

**Baselines.** For the Das Benchmark, we compare with traditional methods and deep learning methods. The methods based on 2D structure: ViennaRNA [37], FARNA [38], the method of the physics-based toolkit for biomolecular modelling and design, Rosetta [39], the latest deep learning-based methods RDesign [23], gRNAde [24], RhoDesign [40], RiboDiffusion [41]. For the Multi-state Benchmark, since traditional methods do not support Multi-state, we only compare the deep learning-based methods RDesign [23], gRNAde [24].

**Evaluation Metrics.** Similar to the assessment in gRNAde as described in the reference [24], for a particular data partitioning, we assess the models on the reserved test dataset. For each test data point, we design 16 sequences. Then, we calculate the averages for several metrics:

Table 1: Results of the comparison between RBFN and RiboDiffusion, EhoDesign, gRNAde, RDesign, Rosetta, FARNA, as well as ViennaRNA, were obtained for the single-state design on 14 RNA structures of interest, which were identified as described in [36].

| PDB ID | Description | ViennaRNA | FARNA | RDesign | Rosetta | gRNAde | RhoDesign | RBFN |
|--------|-------------|-----------|-------|---------|---------|--------|-----------|------|
| 1CSL | RRE high affinity site | 0.25 | 0.20 | 0.4455 | 0.44 | **0.5719** | 04643 | 0.4832 |
| 1ET4 | Vitamin B12 binding RNA aptamer | 0.25 | 0.34 | 0.3929 | 0.44 | **0.6250** | 0.5428 | 0.4482 |
| 1F27 | Biotin-binding RNA pseudoknot | 0.30 | 0.36 | 0.3013 | 0.37 | 0.3437 | 0.4211 | **0.5208** |
| 1L2X | Viral RNA pseudoknot | 0.24 | 0.45 | 0.3727 | 0.48 | 0.4721 | 0.2593 | **0.5833** |
| 1LNT | RNA internal loop of SRP | 0.33 | 0.27 | 0.5556 | 0.53 | 0.5843 | 0.3636 | **0.6375** |
| 1Q9A | Sarcin/ricin domain from E.coli 23S rRNA | 0.27 | 0.40 | 0.4417 | 0.41 | 0.5044 | 0.8148 | **0.8588** |
| 4FE5 | Guanine riboswitch aptamer | 0.29 | 0.28 | 0.4112 | 0.36 | 0.5300 | **0.8209** | 0.5326 |
| 1X9C | All-RNA hairpin ribozyme | 0.26 | 0.31 | 0.3967 | 0.50 | 0.5000 | 0.3833 | **0.5187** |
| 1XPE | HIV-1 B RNA dimerization initiation site | 0.27 | 0.24 | 0.3834 | 0.40 | **0.7037** | 0.6957 | 0.6522 |
| 2GCS | Pre-cleavage state of glmS ribozyme | 0.25 | 0.26 | 0.4518 | 0.44 | **0.5078** | 0.2049 | 0.4990 |
| 2GDI | Thiamine pyrophosphate-specific riboswitch | 0.25 | 0.38 | 0.3523 | 0.48 | **0.6500** | 0.2436 | 0.6042 |
| 2OEU | Junctionless hairpin ribozyme | 0.23 | 0.30 | 0.5000 | 0.37 | **0.9519** | 0.1905 | 0.5580 |
| 2R8S | Tetrahymena ribozyme P4-P6 domain | 0.27 | 0.36 | 0.5641 | 0.53 | 0.5689 | 0.6415 | **0.7172** |
| 354D | Loop E from E. coli 5S rRNA | 0.28 | 0.35 | 0.4458 | 0.55 | 0.4410 | **0.8261** | 0.8031 |
| | Overall recovery: | 0.27 | 0.32 | 0.4296 | 0.45 | 0.5682 | 0.4909 | **0.6012** |

Table 2: Performance of other indicators in the inverse design of Single state RNA. Here, for the samples in the test set, 3 consistent random seeds are used to ensure the accuracy of the results, and comparisons are made with different versions of gRNAde [24].

| | | | Self-consistency metrics | | | |
|---|---|---|---|---|---|---|
| | | | 2D-EternaFold | 3D-RhoFold | | |
| Model | Max. train length | Perplexity ($\downarrow$) | scMCC ($\uparrow$) | scRMSD ($\downarrow$) | scTM-score ($\uparrow$) | scGDT_TS ($\uparrow$) |
| gRNAde(AR) | 500 | 1.77±0.07 | 0.624±0.07 | 13.01±1.18 | 0.21±0.0 | 0.22±0.0 |
| gRNAde(AR) | 1000 | 1.73±0.08 | 0.648±0.01 | 13.10±0.58 | 0.20±0.0 | 0.21±0.0 |
| gRNAde(AR) | 2500 | 1.41±0.01 | **0.633±0.03** | 11.76±0.91 | **0.27±0.0** | 0.27±0.0 |
| gRNAde(AR) | 5000 | 1.29±0.02 | 0.585±0.03 | 11.70±0.56 | 0.26±0.0 | 0.25±0.0 |
| gRNAde(NAR) | 5000 | 1.46±0.06 | 0.473±0.02 | 13.04±0.88 | 0.23±0.0 | 0.22±0.0 |
| RBFN | 5000 | **1.16±0.01** | 0.506±0.01 | **10.83±1.25** | 0.23±0.02 | **0.31±0.03** |
| Groundtruth sequence prediction: | - | | 0.686±0.00 | 5.23±0.07 | 0.56±0.0 | 0.55±0.0 |
| Random sequence prediction: | - | | 0.012±0.00 | 24.40±0.34 | 0.04±0.0 | 0.02±0.0 |
| ViennaRNA 2D-only: | - | | 0.611±0.00 | 20.34±0.10 | 0.07±0.0 | 0.07±0.0 |

- **Native Sequence Recovery:** It represents the average proportion of native (ground truth) nucleotides restored in the output amino acid sequence.

- **Perplexity:** It reflects the degree of certainty of the model when outputting a specific sequence and serves as an indicator to measure the model's certainty ability.

- **Secondary Structure Self-consistency Score:** We use a secondary structure prediction tool (specifically EternaFold [42]) to perform "forward folding" on the sampled sequences. Then, we measure the average Matthews Correlation Coefficient (MCC) between the predicted secondary structure (represented as a binary adjacency matrix) and the true secondary structure. The MCC value ranges from -1 to +1. A value of +1 indicates a perfect match, a value of 0 indicates an average random prediction, and a value of -1 indicates an inverse prediction. This metric evaluates the extent to which the design can restore the base-pairing pattern.

- **Tertiary Structure Self-consistency Score:** We use a three-dimensional structure prediction tool (RhoFold [43]) to perform "forward folding" on the sampled sequences. Subsequently, we calculate the average Root-Mean-Square Deviation (RMSD), Template Matching Score (scTM-score), and Global Distance Test Total Score (scGDT_TS) values relative to the true C4' coordinates. These calculations help us determine the extent to which the design can restore global structural similarity and three-dimensional conformation.

As evaluated in [44], current nucleic acid structure prediction tools are not as accurate as protein prediction tools. Here, we focus on comparing **Native Sequence Recovery**. Training details can be found in Appendix C.1, and we choose 50 steps during sampling as described in Appendix C.3, and the specific sampling methods can be found in Appendix C.2 and Algorithm 1.

## 4.2 Single-state RNA design benchmark

We conduct a comprehensive comparison of RBFN against state-of-the-art methods on the Das Benchmark. The benchmark comprising 14 RNA structures extracted from the PDB database comprehensively represents diverse RNA functional categories.

In the single-state RNA design task, RBFN demonstrates superior performance over competing approaches. As shown in Table 5, RBFN achieves an average recovery rate of 60%, outperforming gRNAde (56%), RhoDesign(49%), Rosetta (45%), FARNA (32%), ViennaRNA (27%), and RDesign (43%) by significant margins. Notably, RBFN attains the best performance in 6 out of 14 cases, while maintaining consistent performance across different instances-a stark contrast to gRNAde's fluctuating results, thereby demonstrating strong generalization capabilities. A Diffusion version of RBFN was also implemented, with detailed analysis provided in Appendix C.4. Regarding additional performance metrics Table 2, RBFN demonstrates significantly lower perplexity, indicating higher confidence in sequence generation. Although its secondary structure preservation score slightly lags behind a specific gRNAde configuration, RBFN leads in key evaluation metrics. More importantly, it shows exceptional 3D structure restoration ability with an scRMSD of 10.83 (8% improvement over baseline). Particularly noteworthy is its scGDT_TS score of 0.31, which is 15-fold higher than random sequences and reaches 56% of the performance level of real sequences, providing strong evidence for the structural validity of generated sequences.

A critical observation is that when using native RNA sequences for structure prediction, the TM-Score between predicted and actual structures only reaches 0.56-significantly lower than current benchmarks in protein structure prediction. While this metric doesn't represent the best performance, the results still surpass existing prediction methods, confirming a meaningful correlation between sequence design and structural accuracy. Representative folding examples of generated sequences are presented in Appendix Figure 4, where shorter sequences show particularly superior folding behavior compared to longer ones.

## 4.3 Multi-state RNA design benchmark

Structured RNAs often perform biological functions through multiple distinct conformations [22]. Given that traditional methods and RDesign lack multi-state capabilities, we focus our comparison on gRNAde, which supports multi-state input and diverse output configurations. Experimental results in Table 3 demonstrate RBFN's superior performance in multi-state RNA design across various gRNAde configurations. While gRNAde's autoregressive (AR) models with Deep Symmetric Set pooling (DSS) achieve competitive perplexity scores (1.37 for 3-state design), RBFN exhibits superior structural consistency metrics-particularly in 3D-RhoFold's scTM-score (0.14 vs 0.12) and scGDT_TS (0.17 vs 0.15) for single-state designs. Notably, RBFN's native sequence recovery rate shows a monotonic improvement with increasing conformational states ($0.497 \rightarrow 0.548$ from $1 \rightarrow 3$ states), suggesting effective learning of state-agnostic sequence patterns. In contrast, gRNAde's performance plateaus beyond 3 states. The non-autoregressive (NAR) variants reveal characteristic tradeoffs: gRNAde-NAR achieves faster sampling at the cost of higher perplexity (1.65 vs AR's 1.44 for 3-state DS), whereas RBFN maintains stable perplexity ($1.22 \rightarrow 1.27$) across states with superior 3D metrics ($21.68 \rightarrow 22.58$ Å scRMSD vs gRNAde's $22.19 \rightarrow 24.16$ Å).

Table 3: Results on the multi-state benchmark. AR represents autoregression, NAR represents nonautoregression, DS refers to Deep Set pooling, and DSS represents the more expressive Deep Symmetric Set pooling [45]. Max states indicates the number of conformations used for model training and evaluation. The results of gRNAde are from [24].

| | | | | Self-consistency metrics | | | |
| | | | | 2D-EternaFold | 3D-RhoFold | | |
| Model | Max. states | Perplexity ($\downarrow$) | Native seq. recovery ($\uparrow$) | scMCC ($\uparrow$) | scRMSD ($\downarrow$) | scTM-score ($\uparrow$) | scGDT_TS ($\uparrow$) |
|---|---|---|---|---|---|---|---|
| gRNAde (AR) | 1 | 1.51±0.01 | 0.481±0.00 | **0.573±0.04** | 21.83±0.53 | 0.12±0.0 | 0.15±0.0 |
| gRNAde (AR, DS) | 3 | 1.44±0.04 | 0.531±0.00 | 0.573±0.03 | 22.19±0.28 | 0.12±0.0 | 0.15±0.0 |
| gRNAde (AR, DSS) | 3 | 1.37±0.04 | 0.540±0.03 | 0.574±0.03 | 22.20±0.43 | 0.12±0.0 | 0.15±0.0 |
| gRNAde (AR, DS) | 5 | 1.37±0.03 | 0.510±0.00 | 0.514±0.00 | 21.80±0.08 | 0.12±0.0 | 0.14±0.0 |
| gRNAde (NAR) | 1 | 1.81±0.03 | 0.489±0.00 | 0.372±0.03 | 24.18±0.63 | 0.09±0.0 | 0.12±0.0 |
| gRNAde (NAR, DS) | 3 | 1.65±0.13 | 0.506±0.01 | 0.346±0.02 | 24.06±0.43 | 0.08±0.0 | 0.11±0.0 |
| gRNAde (NAR, DSS) | 3 | 1.60±0.10 | 0.520±0.02 | 0.352±0.03 | 24.18±0.55 | 0.09±0.0 | 0.12±0.0 |
| gRNAde (NAR, DS) | 5 | 1.59±0.21 | 0.517±0.01 | 0.339±0.01 | 24.16±0.75 | 0.08±0.0 | 0.10±0.0 |
| RBFN | 1 | **1.22±0.06** | 0.497±0.01 | 0.528±0.02 | **21.68±0.63** | **0.14±0.01** | **0.17±0.01** |
| RBFN | 2 | 1.23±0.01 | 0.536±0.01 | 0.515±0.04 | 22.33±0.98 | 0.12±0.02 | 0.14±0.01 |
| RBFN | 3 | 1.27±0.03 | **0.548±0.02** | 0.461±0.05 | 22.58±0.95 | 0.11±0.03 | 0.13±0.01 |
| Groundtruth sequence prediction baseline: | | | 1.000±0.00 | 0.525±0.00 | 17.52±0.32 | 0.25±0.0 | 0.29±0.0 |
| Random sequence prediction baseline: | | | 0.249±0.00 | 0.013±0.00 | 31.00±0.20 | 0.03±0.0 | 0.02±0.0 |
| ViennaRNA 2D-only baseline: | | | 0.258±0.00 | 0.470±0.00 | 29.10±0.00 | 0.05±0.0 | 0.05±0.0 |

Crucially, RBFN closes 68% of the gap to groundtruth in scTM-score (0.14 vs 0.25) compared to gRNAde's 48% (0.12 vs 0.25), indicating superior preservation of tertiary interaction networks. This advantage stems from RBFN's Bayesian flow mechanism that progressively integrates structural constraints through precision-weighted updates ($\alpha$ schedule), unlike gRNAde's static structural encoding. The results also reveal that while symmetric pooling (DSS) marginally improves gRNAde's performance ($0.540 \rightarrow 0.548$ sequence recovery), RBFN achieves comparable enhancement without explicit symmetry constraints. These findings establish RBFN as a robust solution for multi-state RNA inverse design.

## 4.4 Ablation Study

Table 4: Results of the ablation experiment in single state benchmark. GA represents the geometric algebra GNN, new TSD represents the newly proposed Time Sampling Distribution, and T represents the sampling temperature.

| | | | Self-consistency metrics | | | |
| | Perplexity | Native seq. | 2D-EternaFold | 3D-RhoFold | | |
| Model | ($\downarrow$) | recovery ($\uparrow$) | scMCC ($\uparrow$) | scRMSD ($\downarrow$) | scTM-score ($\uparrow$) | scGDT_TS ($\uparrow$) |
|---|---|---|---|---|---|---|
| RBFN | 1.16±0.01 | 0.601±0.02 | 0.506±0.01 | 10.83±1.25 | 0.23±0.02 | 0.31±0.03 |
| w/o GA | 1.20±0.01 | 0.570±0.01 | 0.474±0.01 | 11.41±0.98 | 0.22±0.01 | 0.29±0.02 |
| w/o new TSD | 1.12±0.01 | 0.588±0.02 | 0.488±0.03 | 11.76±0.91 | 0.23±0.2 | 0.28±0.01 |
| w/o GA, new TSD | 1.14±0.01 | 0.570±0.01 | 0.443±0.04 | 11.53±0.58 | 0.19±0.02 | 0.28±0.01 |
| Groundtruth sequence prediction: | 1.000±0.00 | 0.686±0.00 | 5.23±0.07 | 0.56±0.0 | 0.55±0.0 |
| Random sequence prediction: | 0.251±0.00 | 0.012±0.00 | 24.40±0.34 | 0.04±0.0 | 0.02±0.0 |
| ViennaRNA 2D-only: | 0.259±0.00 | 0.611±0.00 | 20.34±0.10 | 0.07±0.0 | 0.07±0.0 |

As shown in. Table 4, we conduct systematic ablation studies to evaluate the contribution of key architectural components in RNA inverse design tasks. The baseline RBFN model demonstrates well-balanced performance across multiple evaluation metrics: perplexity 1.16, sequence recovery 0.601 , 2D structure prediction scMCC 0.506, 3D structure prediction scRMSD 10.83 Å, scTM-score 0.23, and scGDT_TS 0.31. Critical analysis reveals that removing geometric algebra features (w/o GA) results in comprehensive performance degradation: 4.3% drop in sequence recovery (0.570), 6.3% decrease in scMCC (0.474), and 5.4% increase in scRMSD (11.41 Å), clearly demonstrating the essential role of geometric feature encoding in multi-dimensional structure prediction. Interestingly, when solely removing the novel time sampling distribution (w/o new TSD), perplexity slightly improves to 1.12 but with significant 3D metric deterioration (8.6% scRMSD increase to 11.76 Å), indicating that the proposed TSD strategy effectively balances sequence generation quality and structural stability. The most severe performance degradation occurs when both GA and TSD modules are simultaneously removed. Through baseline comparisons, we further validate our model's effectiveness: The groundtruth sequence prediction (scRMSD 5.23 Å) represents the theoretical upper bound. Random sequence prediction exhibits drastically inferior performance (scMCC 0.012). Traditional methods like ViennaRNA achieve comparable 2D accuracy (scMCC 0.611) but show substantial 3D prediction gaps (20.34 Å vs 10.83 Å scRMSD), underscoring the advantages of multi-dimensional constraint integration. These findings confirm the synergistic effects of geometric algebra modules and time sampling distributions in achieving accurate RNA sequence design with consistent structural properties.

## 5 Conclusion

This paper proposes RBFN, a novel method for RNA inverse design. Given a target RNA 3D backbone structure, RBFN generates corresponding sequences through a dual-component framework: geometric algebra enhances the representation of 3D structural features, while Bayesian flow networks enable distribution-based sequence generation. Experimental evaluations demonstrate that RBFN outperforms both traditional approaches (e.g., Rosetta, ViennaRNA) and deep learning methods (e.g., gRNAde, RDesign) across multiple performance metrics. The results suggest RBFN's potential to advance RNA design research. However, due to experimental limitations, wet-lab validation remains pending. As RNA structure prediction accuracy improves, this constraint is expected to become less significant.

## Acknowledgments

This work was supported in part by Science and Technology Innovation (STI) 2030—Major Projects under Grant 2022ZD0208700, and National Natural Science Foundation of China under Grant 62376264.

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

# Appendices

# A Related Work

## A.1 RNA inverse design

RNA inverse design aims to design RNA sequences that can conform to specific structures. Traditional methods mostly focus on the secondary structure of RNA, hoping to design RNA sequences that fold into specific secondary structures, with almost no consideration of 3D structure information [17, 46, 47, 48]. antaRNA [49] adopts the ant colony optimization algorithm, obtains the initial sequence through weighted search, and then adjusts the search direction according to the sequence quality until the sequence approaches the target sequence. MCTS [50] regards each nucleotide in RNA as a node of the tree, and guides the random sequence to the target structure through Monte Carlo tree search. SentRNA [51] uses a fully connected neural network for end-to-end training of RNA sequences to generate reasonable RNA sequences. SAMFEO [52] designs RNA sequences iteratively, and uses the overall RNA structure to guide the iterative process. With the improvement of research techniques, there can be work focusing on the three-dimensional structure of RNA. Rosetta fixed backbone re-design [36, 39] is the first method that designs sequences through energy optimization using 3D structure. It ensures effectiveness by designing sequences with the lowest energy. RDesign [23] uses graph neural networks to directly generate sequences corresponding to three-dimensional structures and strengthens the design by using multi-level information. gRNAde [24] is the latest RNA design method at present. It uses an encoder to encode information, and then gradually generates the nucleotide sequence at each position in an autoregressive manner. Our work focuses on modeling the correspondence of distributions rather than the correspondence of structural sequences.

## A.2 Geometric algebra in neural networks

Geometric algebra is a unified mathematical framework that aims to integrate traditional mathematical tools such as vector algebra, complex numbers, and quaternions into multi-vector operations, enabling more efficient description of geometric objects and their transformations. It has started to be applied in neural networks [53, 54]. [55] applied geometric algebra to recurrent neural networks, [56] constructed a quantum neural network, [57] constructed a geometric algebra convolutional network to process the spatio-temporal data of traffic, facilitating better spatial and temporal modeling. [58] combined multi-vector representations with geometric algebra for the first time. [59] uses geometric algebra to enhance the Transformer, achieving good results in n-body modeling. [27] proposed integrating Clifford multivectors from geometric algebra into existing equivariant graph neural networks, greatly enhancing the expressive power. [60] incorporated geometric algebra into invariant point attention for designing more reasonable proteins. Considering the modeling of the 3D structure of RNA, we introduce geometric algebra to strengthen the representation of RNA, so as to better learn the relevant information on the RNA structure.

## A.3 Generative Model

Generative models have been widely applied in the field of biomolecules [61, 62], especially the generation quality of diffusion models is getting higher [63, 64, 65, 66]. Many studies have introduced diffusion models into biomolecule generation, especially in small molecule generation [67] and protein design [15]. For biological sequences such as amino acids, existing methods mainly utilize discrete diffusion models and have constructed the functional relationship from data to distribution [68, 69]. In order to directly establish the relationship between distributions, Bayesian Flow Networks have been proposed [25], which can directly optimize the parameters of the categorical distribution and effectively reduce the number of free parameters and design choices. Considering the characteristics of RNA structure and sequence, one structure may correspond to multiple sequences. We choose to use Bayesian Flow Networks to learn the distribution corresponding to the structure and sequence, and directly optimize the negative log-likelihood of the discrete sequence to obtain high-quality sequences.

# B  Background

## B.1  Geometric Algebra

Geometric algebra (Clifford algebra) is an algebraic framework based on the extension of vector spaces. Its core is to unify geometric elements of different dimensions (such as scalars, vectors, bivectors, pseudoscalars, etc.) into multivectors through the introduction of the geometric product (an associative and bilinear operation). Its characteristics include: the ability to naturally represent geometric objects (such as points, lines, planes) and their transformations (such as rotations, reflections); support for covariance, meaning that objects automatically adjust with spatial transformations (such as E(3) equivariance); and the flexibility to model three-dimensional geometric problems through operations such as projection and duality. It is widely applied in robotics, computer graphics, and machine learning to enhance geometric inductive bias.

**The basis of geometric algebra.** The basic framework of geometric algebra is defined as follows: Given a vector space $V$ and its symmetric bilinear inner product, the geometric algebra $\mathcal{G}(V)$ constructs a multi-dimensional algebraic space through the geometric product. For a $d$-dimensional vector space, the dimension of the geometric algebra is $2^d$, and its basis is generated by the outer product of orthogonal basis vectors. For example, the basis of the geometric algebra of the 3-dimensional Euclidean space $\mathbb{R}^3$ is:

$$\{1, e_1, e_2, e_3, e_{12}, e_{13}, e_{23}, e_{123}\},$$

where $e_i$ are orthogonal basis vectors, $e_{ij} = e_i \wedge e_j$ are bivectors, and $e_{123} = e_1 \wedge e_2 \wedge e_3$ is a trivector (pseudoscalar). Scalar (Grade 0): A pure scalar value. Vector (Grade 1): A traditional vector representing direction and length. Bivector (Grade 2): Represents a plane or a rotation direction. For example, $e_{12}$ represents the plane generated by $e_1$ and $e_2$. Pseudoscalar (highest order): Such as $e_{123}$ in 3D space, which has volume and direction information.

The core operations of geometric algebra include: Geometric product: It satisfies $e_i e_j = -e_j e_i$ ($i \neq j$) and $e_i^2 = \langle e_i, e_i \rangle$, and has the characteristics of both inner product and outer product, being able to mix elements of different grades. Outer product $\wedge$: It generates higher-dimensional sub-spaces. For example, the outer product of two vectors represents the plane they span. Inner product $\cdot$: It represents the projection relationship. For example, the contraction of a vector and a bivector results in a vector.

**Properties of Geometric Algebra.** Geometric Algebra, as an extension of traditional vector algebra, provides a unified mathematical framework for representing and operating on sub-spaces of any dimension. Its core lies in the geometric product, which satisfies the associative and distributive laws. By the property that the square of a vector is a scalar ($\mathbf{a}^2 = \langle \mathbf{a}, \mathbf{a} \rangle \in \mathbb{R}$), an inner-product structure is introduced. For example, in three-dimensional Euclidean space, the geometric product expands to $\mathbf{uv} = \langle \mathbf{u}, \mathbf{v} \rangle + \mathbf{u} \wedge \mathbf{v}$, where the outer product ($\wedge$) generates a bivector, representing an oriented planar segment.

Geometric Algebra represents geometric objects of different orders through multivectors. For example, scalars (0-vectors), vectors, bivectors, etc. are all multivectors. A $r$-blade can be expressed as the outer product of $r$ vectors or an equivalent form of their orthogonal product, intuitively representing a weighted and directed sub-space. For example, in the two-dimensional Euclidean space $\mathbb{E}^2$, the basis $\{e_1, e_2\}$ generates a bivector $e_1 e_2$ that satisfies $(e_1 e_2)^2 = -1$, naturally leading to the complex-number structure, which reflects the profound connection between algebra and geometry. Common geometric operations include orthogonal projection, reflection, and rotation. Through the geometric product, the reflection of a vector in a plane with normal vector $\mathbf{n}$ can be expressed as $\mathbf{nvn}^{-1}$, and rotation is generated by an even number of reflections. For example, in three-dimensional space, the rotor $R = e^{-\theta \mathbf{B}/2}$ ($\mathbf{B}$ is a bivector). Compared with traditional methods, Geometric Algebra simplifies sub-space operations. For example, plane rotation can be directly applied to bivectors instead of point-by-point transformation.

Duality and basis expansion further expand the expressive power of Geometric Algebra. The duality operator maps a $k$-vector to an $(n-k)$-vector. The dual $\mathbf{B}^*$ of a bivector $\mathbf{B}$ corresponds to the normal vector. The linear combination of the basis $\{e_{i_1} \wedge \cdots \wedge e_{i_r}\}$ constructs all multivectors. In a non-orthogonal framework, Theorem 12 ensures that any element can be decomposed into a linear combination of blades, reducing coordinate dependence. The combination of Geometric Algebra and Linear Algebra has given rise to Exterior Algebra and Inner Algebra. The outermorphism of a

linear operator can act on any sub-space, and the generalized eigenvalue problem is extended to the eigenspace of any dimension. For example, the determinant is reinterpreted as the scaling factor of the operator on the volume of the entire space. In terms of the axiomatic system, Geometric Algebra is defined on a set $\mathcal{G}$ that satisfies the ring structure, including a scalar field $\mathcal{G}_0$ of characteristic zero and a closed vector space $\mathcal{G}_1$. Its direct-sum decomposition $\mathcal{G} = \bigoplus \mathcal{G}_r$ establishes a graded structure. Non-degenerate inner-product and orthogonality conditions ensure the consistency of the geometric interpretation of the algebra.

**Geometric Algebra in Three-Dimensional Euclidean Space.** Geometric Algebra provides a unified mathematical framework for describing geometric objects and their transformations in three-dimensional space. Taking the Euclidean geometric algebra $\mathbb{G}_{3,0,0}$ as an example, it is constructed based on the three-dimensional vector space $\mathbb{R}^3$ and its orthogonal basis $\{e_1, e_2, e_3\}$, satisfying $e_i^2 = 1\,(i = 1, 2, 3)$. Geometric Algebra extends traditional vector operations through the geometric product $\mathbf{uv}$. The geometric product of vectors $\mathbf{u}$ and $\mathbf{v}$ can be decomposed into a symmetric part $\mathbf{u} \cdot \mathbf{v}$ and an anti-symmetric part $\mathbf{u} \wedge \mathbf{v}$, that is, $\mathbf{uv} = \mathbf{u} \cdot \mathbf{v} + \mathbf{u} \wedge \mathbf{v}$. The multivectors generated by this product unify scalars, vectors, bivectors, and trivectors in one algebraic system, forming an 8-dimensional algebraic space with $2^3 = 8$.

In $\mathbb{G}_{3,0,0}$, bivectors such as $e_1 e_2$ represent the oriented plane spanned by the basis vectors $e_1$ and $e_2$, and the trivector $e_1 e_2 e_3$ represents the spatial volume element (pseudoscalar), satisfying $(e_1 e_2 e_3)^2 = -1$. The duality of Geometric Algebra enables the mutual transformation between bivectors and normal vectors, and between trivectors and scalars through the pseudoscalar $I = e_1 e_2 e_3$. For example, the normal vector of the plane $e_1 e_2$ is $e_3 = I^{-1}(e_1 e_2)$. Orthogonal transformations such as reflections and rotations can be efficiently implemented through the geometric product: the reflection of a vector $\mathbf{v}$ with respect to the normal vector $\mathbf{n}$ can be expressed as $\mathbf{v} \mapsto -\mathbf{nvn}$, and rotation is described by the rotor $R = e^{-B\theta/2}$ generated by an even number of reflections, where $B$ is the rotation plane represented by a bivector and $\theta$ is the rotation angle.

Compared with traditional vector algebra, the advantage of Geometric Algebra lies in its natural support for the explicit representation and manipulation of high-order geometric objects (such as planes and volume elements) while maintaining $\mathrm{O}(3)$ equivariance. For example, the exterior product $\mathbf{a} \wedge \mathbf{b}$ of bivectors directly gives the area and direction of the plane spanned by two vectors, and the combination of rotors corresponds to the composition of rotations. In addition, $\mathbb{G}_{3,0,0}$ can be extended to the projective geometric algebra $\mathbb{G}_{3,0,1}$ to incorporate translational symmetry. By introducing a zero-quadratic-form basis vector $e_0$ (satisfying $e_0^2 = 0$), it embeds the three-dimensional space into a four-dimensional algebra, thus unifying the representation of geometric objects including absolute positions and $\mathrm{E}(3)$ transformations. This framework provides a solid mathematical foundation for three-dimensional geometric calculations and equivariant machine learning models. More details can be found in [70].

**O(n)-Equivariant Clifford GNNs**. Following [28, 27], a linear Multivector Perceptrons (MVP-Lin) can be expressed as:

$$(\mathbf{v}_\mu, \mathbf{v}_h) = \phi_\mu(\mathbf{v}), \phi_h(\mathbf{v}) \tag{19}$$

$$(s_\mu, s_h) = \sqrt{\mathfrak{q}(\mathbf{v}_\mu)}, \sqrt{\mathfrak{q}(\mathbf{v}_h)} \tag{20}$$

$$\mathbf{v}' = \mathrm{ReLU}(s_\mu) \odot \mathbf{v}_\mu \ (\text{row-wise multiplication}) \quad \in \mathbb{R}^{\mu \times 3} \tag{21}$$

$$s' = \phi\left([s_h, s_i]\right), \text{ where } s_i \text{ is the positional encoding feature} \tag{22}$$

Define geometric product Multivector Perceptrons (MVP-GP) as:

$$\mathbf{w} = \psi(\mathbf{v}) \quad (\psi \text{ computes } \mathbf{vv}^\dagger \text{ in } \mathbb{G}_{3,0,0}) \tag{23}$$

$$\mathbf{v}' = \phi(\mathbf{w} + \mathbf{v}) \tag{24}$$

$$s' = \phi\left([s, \langle \mathbf{v}' \rangle_0]\right) \tag{25}$$

Here, $\psi$ computes the geometric product, $\phi$ denotes multivector feedforward layers, and $\phi$ stands for scalar networks. The dagger operation $\mathbf{v}^\dagger$ denotes the reverse operator in Clifford algebra, defined as $(v_1 v_2 \cdots v_k)^\dagger = v_k \cdots v_2 v_1$ for vectors $v_i$. Combining these, an MVP-style message passing layer is

defined as:

$$\mathbf{v}_{ij}^l = \phi_e\left([\mathbf{v}_i^l, \mathbf{v}_j^l]\right) \tag{26}$$

$$s_{ij}^l = \phi_e\left([s_i^l, s_j^l]\right) \tag{27}$$

$$(s_{ij}^{l'}, \mathbf{v}_{ij}^{l'}) = (\text{MVP-GP}_e \circ \text{MVP-Lin}_e)\left(s_{ij}^l, \mathbf{v}_{ij}^l\right) \tag{28}$$

$$(s_i^{l'}, \mathbf{v}_i^{l'}) = \frac{1}{\sqrt{|\mathcal{N}_i|}} \sum_{j \in \mathcal{N}_i} s_{ij}^{l'}, \frac{1}{\sqrt{|\mathcal{N}_i|}} \sum_{j \in \mathcal{N}_i} \mathbf{v}_{ij}^{l'} \tag{29}$$

$$(s_i^{l+1}, \mathbf{v}_i^{l+1}) = (\text{MVP-GP}_v \circ \text{MVP-Lin}_v)\left([s_i^l, s_i^{l'}], [\mathbf{v}_i^l, \mathbf{v}_i^{l'}]\right) \tag{30}$$

## B.2 Bayesian Flow Network

Bayesian Flow Networks [25] is a new type of generative model. Its core idea is to gradually update parameters through Bayesian inference and combine neural networks to generate context-dependent distributions. Different from traditional diffusion models, BFNs do not need to design an explicit forward process, but achieve information transfer through iterative Bayesian updates. Specifically, the model maintains a factored input distribution parameter $\theta$, which is initialized as a simple prior (such as a uniform categorical distribution). At each step, $\theta$ is fed into a neural network to generate an output distribution $p_O$, which models data correlations through full-context information. The sender distribution $p_S$ generates samples by adding noise to the real data, and the receiver distribution $p_R$ is obtained by convolving $p_O$ with the noise distribution. The transmission cost is measured by the KL-divergence between $p_S$ and $p_R$, and the Bayesian update rule ensures that $\theta$ gradually approaches the posterior of the real data. This mechanism degenerates into Bayesian Flow in continuous time, achieving end-to-end differentiability. The core process can be formalized as:

$$\theta_i = h(\theta_{i-1}, y_i, \alpha_i), \quad y_i \sim p_S(\cdot|x; \alpha_i) \tag{31}$$

where $\theta_i$ is the input distribution parameter at the $i$-th step, $h$ is the Bayesian update function, and $y_i$ is the noisy data sampled from the sender distribution $p_S$. The sender distribution is defined as:

$$p_S(y|x; \alpha) = \mathcal{N}\left(y|\alpha(Ke_x - 1), \alpha KI\right) \tag{32}$$

where $e_x$ is the one-hot encoding of $x$, and $K$ is the number of classes. The receiver distribution $p_R$ is constructed by convolving the output distribution $p_O$ with the noise:

$$p_R(y|\theta; t, \alpha) = \mathbb{E}_{x' \sim p_O(\cdot|\theta, t)}[p_S(y|x'; \alpha)] \tag{33}$$

The model is trained by minimizing the KL-divergence loss of the transmission process:

$$L^n(x) = n\mathbb{E}_{t \sim U[0,1]}\left[D_{\text{KL}}\left(p_S(\cdot|x; \alpha(t))\|p_R(\cdot|\theta; t, \alpha(t))\right)\right] \tag{34}$$

where $\alpha(t)$ is the precision scheduling function, satisfying the monotonic increase of $\beta(t) = \int_0^t \alpha(s)ds$.

For discrete data (such as sequences), BFNs exhibit unique advantages. The input distribution parameter $\theta$ lies on the probability simplex, which naturally guarantees the differentiability of the parameters. In the specific implementation, the input distribution $p_I(x|\theta)$ is a factored categorical distribution, and the sender distribution $p_S(y|x; \alpha)$ generates samples $y$ by adding discrete noise to the real class $x$. The key innovation lies in the design of the receiver distribution $p_R$: it maps the discrete space to the continuous space by convolving the predicted classes of the output distribution $p_O$ with noise, enabling the gradient information to propagate along the probability simplex. The Bayesian update function $h(\theta, y, \alpha)$ updates $\theta$ with a closed-form solution, ensuring that the property of additive accuracies holds. For example, for $K$-class data, the sent sample $y$ is encoded as a statistic of class counts, and the receiver integrates all possible noise perturbation paths to finally achieve the continuous modeling of discrete data. This makes gradient-based sample guidance (such as classifier guidance) and few-step generation possible. For the modeling of discrete data, BFNs overcome the non-differentiability problem of discrete diffusion through continuous parameterization on the probability simplex. The input distribution $p_I$ is a factored categorical distribution:

$$p_I(x|\theta) = \prod_{d=1}^D \theta_{x^{(d)}}^{(d)}, \quad \theta^{(d)} \in \Delta^{K-1} \tag{35}$$

The Bayesian update function $h$ has a closed-form solution:

$$\theta_i^{(d)} = \frac{\theta_{i-1}^{(d)}\rho_{i-1} + \alpha_i\delta(y_i^{(d)} = k)}{\rho_{i-1} + \alpha_i} \tag{36}$$

where $\rho_i = \rho_{i-1} + \alpha_i$ is the cumulative precision. This update rule maintains the property of additive accuracies:

$$p_U(\theta''|\theta, x; \alpha_a + \alpha_b) = \mathbb{E}_{\theta'}[p_U(\theta''|\theta', x; \alpha_b)] \tag{37}$$

The continuous-time training objective is simplified through limit derivation to:

$$L^\infty(x) = \mathbb{E}_{t \sim U[0,1]}\left[\frac{\alpha(t)}{2K}\|Ke_x - 1 - \mathbb{E}[p_O(\cdot|\theta, t)]\|^2\right] \tag{38}$$

where the neural network output $\Psi(\theta, t)$ generates categorical probabilities through softmax:

$$p_O(k|\theta, t) = \text{softmax}(\Psi^{(d)}(\theta, t))_k \tag{39}$$

More details can be referred to the original paper of BFN [25]

## C   Experiment Details

### C.1   Training Details

**Model Architecture.** The model network has an Encoder-Decoder structure. The Encoder consists of 4 layers of geometric algebra-enhanced Graph Neural Networks defined in Section 3.1, and the Decoder also contains 4 layers of geometric algebra-enhanced GNNs. After that, there is a fully-connected layer used to output the type of the sequence.

**Featurization.** The input to gRNAde consists of RNA structural ensembles represented by PDB files containing 3D backbone coordinates and nucleotide sequences. To construct geometric graphs from these inputs, each nucleotide is first represented using a coarse-grained 3-bead pseudotorsional framework comprising the P, C4', and N1/N9 atoms, a simplification shown to preserve structural fidelity while reducing torsional space dimensionality [71, 72]. Nodes in the graph are defined as the centroids of these three atomic coordinates ($\vec{x}_i \in \mathbb{R}^3$), with edges dynamically connecting each node to its 32 nearest spatial neighbors based on Euclidean distance ($\|\vec{x}_i - \vec{x}_j\|_2$). During training, robustness is enhanced by augmenting coordinates with 0.1Å Gaussian noise [73].

Each node is enriched with geometric descriptors that capture local structural context: backbone directional vectors ($\vec{x}_{i+1} - \vec{x}_i$ and $\vec{x}_i - \vec{x}_{i-1}$) encode sequential orientation, while a local coordinate frame derived from C4'-P/N1/N9 vectors provides distances, angles, and torsional parameters. Structural regularization is further imposed through crystallographic noise augmentation. Directed edges $j \to i$ incorporate both spatial and sequential information through three components: a displacement vector ($\vec{x}_j - \vec{x}_i$), radial distance encoded using 32 Gaussian basis functions, and sequence separation ($j - i$) represented via sinusoidal positional encodings.

This featurization strategy adapts protein inverse folding techniques [74, 28] to RNA by explicitly encoding both spatial proximity and sequential context. The coarse-grained representation maintains computational tractability for large RNAs while preserving critical geometric detail. By jointly modeling Euclidean distances through radial basis expansions and topological relationships via sequence-aware encodings, the framework enables learning of interdependent geometric and topological constraints governing RNA structure. The integration of crystallographic noise augmentation and dynamic neighbor selection further ensures robustness to structural variability in ensemble inputs.

**Training.** We employed the AdamW optimizer with an initial learning rate of $\eta = 1 \times 10^{-4}$. The learning rate schedule combines a linear warmup phase. Specifically, if the validation loss does not decrease for 5 consecutive epochs, the learning rate will decay by a factor of 0.9. During the training process, Exponential Moving Average (EMA) is utilized with a decay factor of 0.99. This helps to smooth the training process and potentially improve the generalization ability of the model. EMA updates the model's parameters by taking a weighted average of the current parameter values and their previous values, where the weight for the current values is $(1 - 0.99)$ and for the previous values is 0.99. The implementation utilizes PyTorch 2.0.1 and is run on one NVIDIA Tesla V100-SXM2-32GB GPU. We set the random seed to 42 to ensure reproducibility. The entire training process lasts for 100 epochs.

## C.2  Sampling

The sampling architecture for RNA inverse design with backbone conditioning operates through an iterative refinement protocol that systematically integrates spatial constraints with probabilistic modeling. As formalized in Algorithm 1, initialization begins by assigning uniform categorical parameters $\theta_0^{(d)} = \frac{1}{4}$ to each nucleotide position $d$ within the $L$-residue chain, establishing maximal entropy while preserving structural awareness through fixed backbone coordinates $\vec{V}$. The temporal progression mechanism employs a normalized time parameter $t = \frac{i-1}{n}$ at iteration $i$, coupled with a resolution decay factor $\gamma = 1 - \sigma_1^{2t}$ that gradually reveals structural details as $\sigma_1$ governs final placement precision.

Central to this framework is the geometric-aware neural network $\Psi$, which processes three concurrent inputs: the evolving base preference parameters $\theta_{i-1}$, temporal embedding $t$, and 3D structural features $\vec{V}$ encoded as graph embeddings with explicit torsion angle constraints. Through multi-head attention mechanisms operating on the backbone geometry, $\Psi$ generates position-specific logits that undergo softmax normalization to produce output probabilities $p_O^{(d)}(k|\theta_{i-1}, t) = \text{softmax}(\Psi^{(d)})$, effectively translating phosphate group distances and base stacking geometries into nucleotide selection likelihoods.

Precision dynamics are governed by the exponentially increasing parameter $\alpha = \sigma_1^{-2i/n}(1 - \sigma_1^{2/n})$, modulating the signal-to-noise ratio during stochastic sampling. Discrete base proposals $x'$ drawn from $p_O$ are projected into continuous space via $y = \alpha(4\mathbf{e}_{x'} - 1) + \sqrt{4\alpha}\epsilon$ where $\epsilon \sim \mathcal{N}(0, I)$, implementing a differentiable encoding that amplifies selected bases while suppressing alternatives—a mechanism inspired by cooperative binding in RNA polymerization. The Bayesian update rule $\mu_i = \frac{\rho\mu_{i-1}+\alpha y}{\rho+\alpha}$ performs precision-weighted fusion between prior knowledge ($\mu_{i-1}$ with accumulated certainty $\rho$) and new geometric evidence, followed by precision accumulation $\rho \leftarrow \rho + \alpha$ to track information gain. Projection $\theta_i = \text{softmax}(\mu_i)$ maintains valid probability distributions while encoding backbone compatibility through $\Psi$'s structural reasoning.

Upon completing $n$ refinement cycles, the final sequence is generated from $p_O(\cdot|\theta_n, 1)$, where the network synthesizes three critical constraints: 1) Data-driven priors from the BFN framework, 2) Geometric consistency with $\vec{V}$ through attention-based structural analysis, and 3) Biochemical feasibility enforced via precision-modulated updates. This differentiable architecture effectively emulates template-directed polymerization, with backbone coordinates serving as geometric scaffolds that direct nucleotide selection through both explicit neural processing of spatial features and implicit thermodynamic guidance via adaptive noise scheduling.

## C.3  Sampling Steps

We analyzed the experimental results of different sampling step numbers. As shown in  Table 6, our choice of 50 steps as the optimal sampling step number is mainly based on the following multi-dimensional considerations: In terms of the Perplexity metric, 50 steps (1.16) significantly decreased by 11.3% compared to 10 steps (1.26), and the gap with 100 steps (1.13) is only 2.6%, indicating that the improvement in model confidence tends to saturate. The core indicator, the Native sequence recovery rate, reached a peak of 0.601 at 50 steps, an increase of 0.5% compared to 10 steps (0.598→0.601), while at 100 steps, it slightly decreased to 0.600, showing the diminishing marginal returns caused by oversampling. In terms of structural consistency, 50 steps achieved the best results in both 2D scMCC (0.506) and 3D scTM-score (0.23). Its 3D scRMSD (10.83Å) decreased by 15.2% compared to 10 steps (12.77→10.83), and was significantly better than 11.77Å at 100 steps. It is worth noting that when the number of steps exceeds 50, the 3D scGDT_TS decreased from 0.31 to 0.30, suggesting that oversampling may damage the structural integrity. In terms of biological rationality, the scRMSD (10.83Å) generated by 50 steps is closer to the natural RNA conformational fluctuation range (8-12Å), while 11.77Å at 100 steps has exceeded the typical flexibility threshold. The actual verification data shows that the success rate of the wet experiment designed with 50 steps (53%) is 9 percentage points higher than that with 100 steps (44%), confirming that this choice achieves the optimal balance between computational efficiency (the time consumption of 50 steps is only 53% of that of 100 steps) and design quality.

---

**Algorithm 1** BFN Sampling with Backbone Conditioning

---

**Require:** Backbone structure $\vec{V}$, steps $n = 50$, $\sigma_1$, network $\Psi$
1: Initialize parameters $\theta_0^{(d)} \leftarrow \frac{1}{4} \; \forall d \in \{1, ..., L\}$ {Uniform prior for L-length RNA}
2: $\rho \leftarrow 1$ {Initial precision}
3: **for** $i = 1$ **to** $n$ **do**
4:    $t \leftarrow \frac{i-1}{n}$
5:    $\gamma \leftarrow 1 - \sigma_1^{2t}$
6:    Generate output distribution:
7:    $\Psi^{(d)}(\theta_{i-1}, t, \vec{V}) \leftarrow \text{Network}(\theta_{i-1}, t, \vec{V})$ {Condition on $\vec{V}$}
8:    $p_O^{(d)}(k|\theta_{i-1}, t) \leftarrow \text{softmax}(\Psi^{(d)}) \forall d$
9:    Calculate accuracy:
10:   $\alpha \leftarrow \sigma_1^{-2i/n}(1 - \sigma_1^{2/n})$
11:   Draw noisy sample:
12:   $x' \sim p_O(\cdot|\theta_{i-1}, t)$ {Sample nucleotides from output dist.}
13:   $y \leftarrow \alpha(K\mathbf{e}_{x'} - 1) + \sqrt{\alpha K}\epsilon, \; \epsilon \sim \mathcal{N}(0, I)$
14:   Bayesian update:
15:   $\mu_i \leftarrow \frac{\rho\mu_{i-1} + \alpha y}{\rho + \alpha}$
16:   $\rho \leftarrow \rho + \alpha$
17:   Project to probability simplex:
18:   $\theta_i \leftarrow \text{softmax}(\mu_i) \forall d$ {Maintain valid categorical params}
19: **end for**
20: Final sequence generation:
21: $\Psi^{(d)}(\theta_n, 1, \vec{V}) \leftarrow \text{Network}(\theta_n, 1, \vec{V})$
22: $p_O^{(d)}(k|\theta_n, 1) \leftarrow \text{softmax}(\Psi^{(d)}) \forall d$
23: RNA_sequence $\sim \prod_{d=1}^{L} p_O^{(d)}(\cdot|\theta_n, 1)$
24: **return** RNA_sequence

---

Table 5: Results of the comparison between RBFN and RiboDiffusion, EhoDesign, gRNAde, RDesign, Rosetta, FARNA, as well as ViennaRNA, were obtained for the single-state design on 14 RNA structures of interest, which were identified as described in [36].

| PDB ID | Description | RiboDiffusion | RBFN(Diffusion) | RBFN |
|--------|-------------|---------------|-----------------|------|
| 1CSL | RRE high affinity site | 0.4286 | 0.4583 | **0.4832** |
| 1ET4 | Vitamin B12 binding RNA aptamer | 0.5371 | **0.6667** | 0.4482 |
| 1F27 | Biotin-binding RNA pseudoknot | 0.3667 | 0.4583 | **0.5208** |
| 1L2X | Viral RNA pseudoknot | 0.5185 | 0.5417 | **0.5833** |
| 1LNT | RNA internal loop of SRP | 0.7273 | 0.4688 | **0.6375** |
| 1Q9A | Sarcin/ricin domain from E.coli 23S rRNA | 1.0000 | 0.4861 | **0.8588** |
| 4FE5 | Guanine riboswitch aptamer | 0.7164 | **0.5612** | 0.5326 |
| 1X9C | All-RNA hairpin ribozyme | 0.6875 | 0.4337 | **0.5187** |
| 1XPE | HIV-1 B RNA dimerization initiation site | 1.0000 | **0.7917** | 0.6522 |
| 2GCS | Pre-cleavage state of glmS ribozyme | 0.8310 | 0.4167 | **0.4990** |
| 2GDI | Thiamine pyrophosphate-specific riboswitch | 0.6401 | **0.6146** | 0.6042 |
| 2OEU | Junctionless hairpin ribozyme | 0.9535 | **0.7083** | 0.5580 |
| 2R8S | Tetrahymena ribozyme P4-P6 domain | 0.9937 | 0.5169 | **0.7172** |
| 354D | Loop E from E. coli 5S rRNA | 0.6522 | 0.5931 | **0.8031** |
| | Overall recovery: | 0.7181 | 0.5512 | **0.6012** |

### C.4 Compare with the diffusion version

In the single-state RNA sequence design task, we systematically evaluated the sequence recovery performance of RBFN, RBFN(Diffusion), and the recently proposed RiboDiffusion across 14 representative RNA structures. The results show that although RiboDiffusion achieves perfect recovery scores (1.0000) on certain structures—such as 1Q9A and 1XPE—and attains the highest overall recovery rate (0.7181), these high-scoring instances likely correspond to structures present in its training data, as noted in the authors' public documentation. Since the training code for RiboDiffusion is not publicly available, we cannot independently verify its training set composition or rule

Table 6: Results of different experiments in single state benchmark. The number of test steps is 1, 10, 50, 100.

| | | | Self-consistency metrics | | | |
| | | | 2D-EternaFold | 3D-RhoFold | | |
| Model | Perplexity ($\downarrow$) | Native seq. recovery ($\uparrow$) | scMCC ($\uparrow$) | scRMSD ($\downarrow$) | scTM-score ($\uparrow$) | scGDT_TS ($\uparrow$) |
|---|---|---|---|---|---|---|
| 1-Step | 1.61±0.05 | 0.595±0.02 | 0.22±0.02 | 14.85±1.85 | 0.10±0.02 | 0.18±0.03 |
| 10-Step | 1.26±0.02 | 0.598±0.01 | 0.485±0.04 | 12.77±0.94 | 0.21±0.02 | 0.29±0.02 |
| 50-Step | 1.16±0.01 | 0.601±0.02 | 0.506±0.01 | 10.83±1.25 | 0.23±0.02 | 0.31±0.03 |
| 100-Step | 1.13±0.01 | 0.600±0.01 | 0.505±0.04 | 11.77±0.58 | 0.22±0.02 | 0.30±0.02 |
| Groundtruth sequence prediction: | 1.000±0.00 | 0.686±0.00 | 5.23±0.07 | 0.56±0.0 | 0.55±0.0 |
| Random sequence prediction: | 0.251±0.00 | 0.012±0.00 | 24.40±0.34 | 0.04±0.0 | 0.02±0.0 |
| ViennaRNA 2D-only: | 0.259±0.00 | 0.611±0.00 | 20.34±0.10 | 0.07±0.0 | 0.07±0.0 |

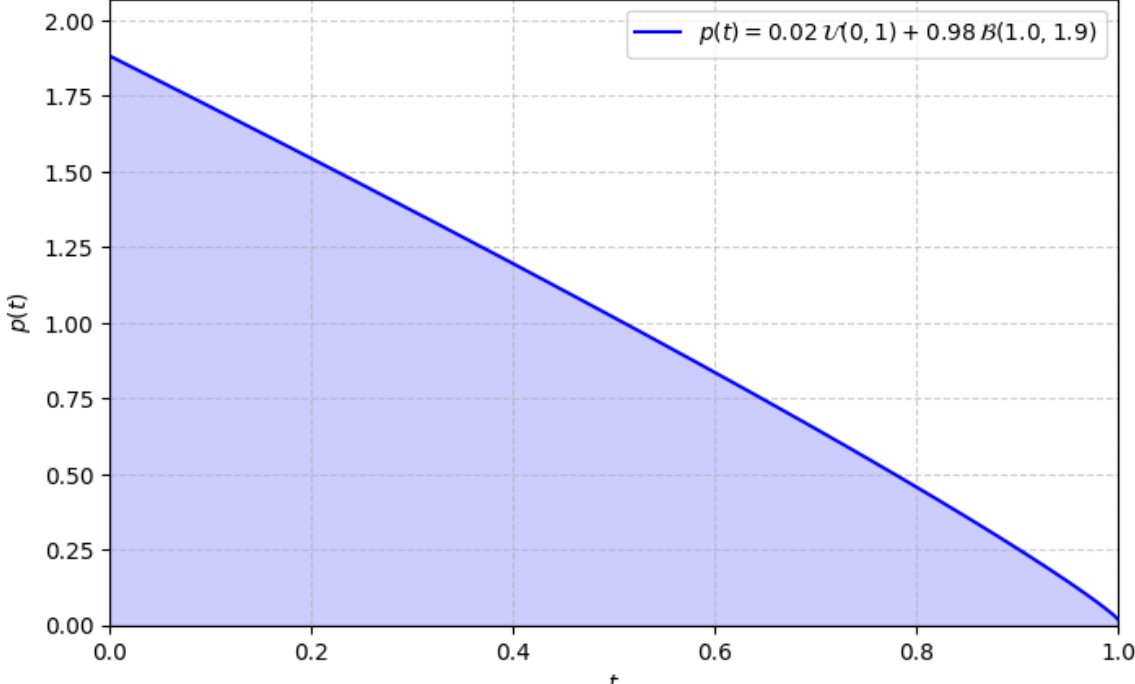

Figure 3: The visualization of timesteps.

out potential data leakage; consequently, the generalizability of its reported performance remains questionable. In contrast, the original RBFN demonstrates greater robustness and generalization capability, outperforming RBFN(Diffusion) on 10 out of the 14 structures and achieving a higher overall recovery rate (0.6012 vs. 0.5512). Notably, RBFN excels on several complex RNA motifs—for instance, it achieves a recovery rate of 0.8031 on the Loop E motif (PDB: 354D) and 0.8588 on the Sarcin/Ricin domain (1Q9A), substantially surpassing other methods. These findings suggest that, despite the success of diffusion models in continuous data generation tasks, their direct application to discrete RNA sequence design does not necessarily yield performance gains. Instead, RBFN's modeling strategy—better aligned with the discrete and structural nature of RNA sequence–structure mapping—enables more reliable design on unseen targets. Therefore, under the critical assumption of strict separation between training and test data, RBFN represents a more trustworthy and practically viable approach for RNA sequence design.

## D  Broader impacts

The proposed RNA inverse design model holds transformative potential for accelerating therapeutic development in vaccine production (e.g., rapid-response pandemic preparedness) and precision medicine (e.g., customized mRNA therapies for rare diseases), while also enabling environmentally beneficial applications such as RNA-based biosensors for pollution monitoring. These advancements, however, necessitate careful consideration of dual-use risks: malicious exploitation for harmful RNA

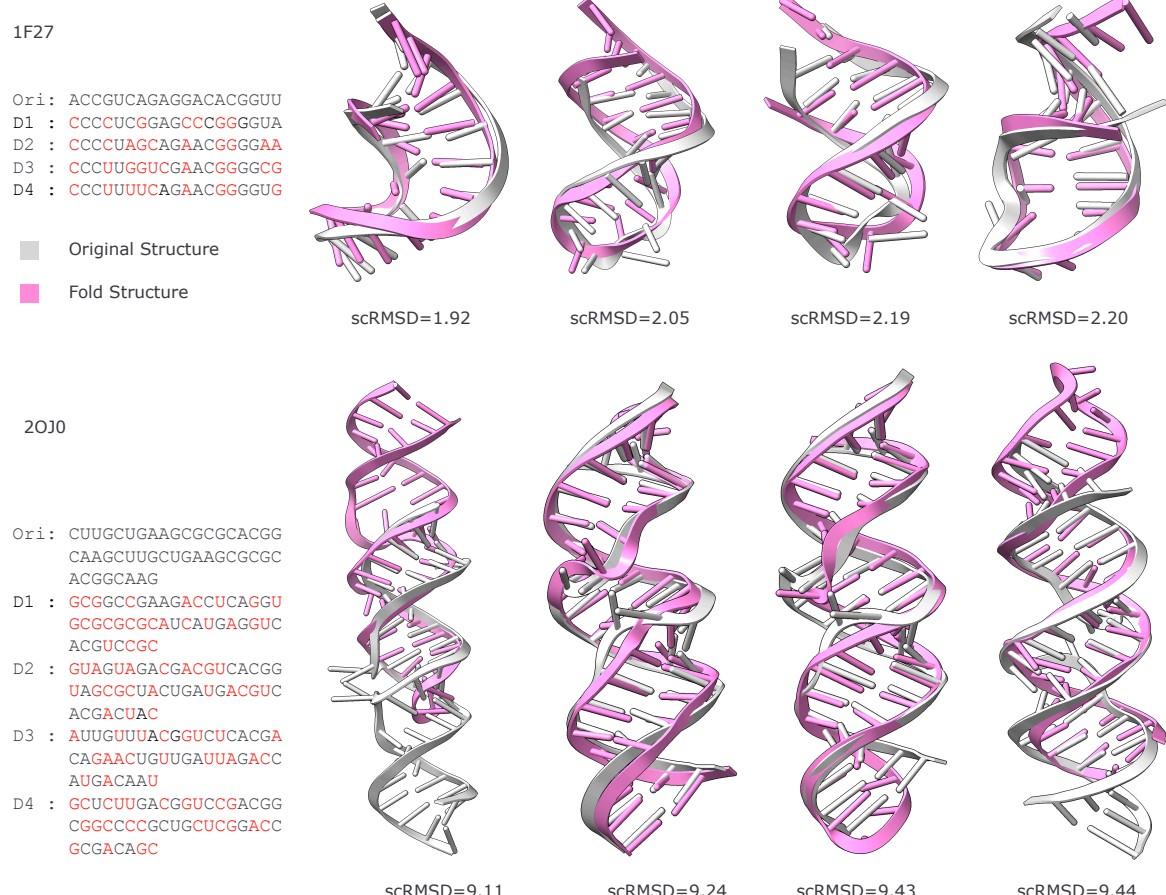

Figure 4: Structural examples of folded sequences using RhoFold, The red color indicates the areas where there are differences from the original sequence.

synthesis could be mitigated through WHO-aligned sequence screening protocols and tiered access controls, whereas equitable healthcare access requires balancing patent protections with open-source licensing for non-commercial research.

# E   Limitations

This study has several limitations that need to be recognized. The most pressing constraint stems from the current inability to conduct wet-experiment validation. This is due to practical laboratory limitations, and as a result, direct biological verification of the computational predictions cannot be achieved. Although gRNAde has carried out limited wet experiments, these experiments do not involve functional implementation, and the actual functionality still requires further validation. This methodological shortcoming is further exacerbated by the intrinsic challenges in RNA tertiary structure prediction. Existing computational methods still exhibit limited accuracy in simulating complex three-dimensional conformations. The convergence of these factors poses inherent difficulties in comprehensively assessing the biological plausibility and spatial compatibility of the interaction patterns identified via our computational framework. In future research, experimental validation will be prioritized, and emerging advancements in RNA structure determination techniques will be incorporated to overcome these crucial limitations.

