# OpenReview forum: "Geometric Algebra-Enhanced Bayesian Flow Network for RNA Inverse Design"
_NeurIPS.cc/2025/Conference — NeurIPS 2025 poster_

### Official Review · Reviewer_LyRL · 2025-06-20

**Clarity:** 3
**Significance:** 3
**Originality:** 3
**Rating:** 5
**Confidence:** 4

**Summary:**

This paper utilized geometric algebra to represent the 3D structure of RNA. Then they proposed a Geometric Algebra-Enhanced Bayesian Flow Network (RBFN) for RNA inverse design. RBFN improves RNA sequence generation by combining geometric algebra for better 3D structure modeling with a Bayesian Flow Network to sample sequences that correspond to a given RNA 3D backbone. The method outperforms existing RNA design tools in benchmarks, showing improvements in sequence recovery and structural consistency.

**Questions:**

- Does the method work for more dynamic RNA structures?
- What are the computational costs for large-scale RNA design tasks?

**Ethical Concerns:**

["NO or VERY MINOR ethics concerns only"]

**Final Justification:**

This method seems solid and useful, so I will remain my score of 5.

**Limitations:**

- Computational Cost: The method could be computationally expensive for large RNA molecules.
- Data Quality Dependence: The method relies on accurate 3D RNA structural data, which may not always be available.
- Lack of Wet-Lab Testing: Without experimental validation, the real-world applicability is uncertain.

**Quality:**

3

**Strengths And Weaknesses:**

Strengths:
- Originality: Combining geometric algebra with Bayesian Flow Networks for RNA design is novel.
- Significance: It has potential applications in RNA-based therapies and synthetic biology.
- Experimental Validation: The method shows superior performance in benchmark tests. The structure evaluation is very good, which may be ignored by previous works.
- Clarity: The paper is well-structured and clear for a technical audience.

Weaknesses:
- Complexity: The technical concepts, particularly geometric algebra and Bayesian Flow Networks, may be challenging for some readers.
- Data Scarcity: The method depends on the availability of high-quality RNA structural data.
- Limited Wet-Lab Validation: There’s no real-world experimental validation of the method.

---

> ### Author Rebuttal · Authors · 2025-07-29
>
> Dear reviewer LyRL,
>
> We express our gratitude for your diligent review and the valuable efforts you have made to assist us in enhancing the manuscript.
>
> > **W1: Complexity: The technical concepts, particularly geometric algebra and Bayesian Flow Networks, may be challenging for some readers.**
>
> We understand the reviewer's concerns about the complexity of Geometric Algebra (GA) and Bayesian Flow Networks (BFN). To lower the barrier to understanding, we will take the following measures in the revised manuscript to ensure that the technical details are clearer and more comprehensible:
>
> **1. Strengthening and layered explanation of the background section**
>
> We will introduce the core concepts of Geometric Algebra in a layered manner, including the basic framework, core operations, practical usage examples, and provide sufficient references for further learning. Additionally, we will further clarify the mechanisms of BFN, including the evolution of continuous-time distributions, differences from traditional diffusion models, handling of discrete data, and visual diagrams of time-step sampling to make it clearer for the readers.
>
> **2. Enhancing the writing of the paper**
>
> We will clarify the descriptions of symbols and related elements, and unify the writing style. We have also written pseudocode in the appendix for reference, and we will add tables explaining the relevant symbols to make it easier for everyone to understand.
>
> > **W2 & L2: Data Scarcity: The method depends on the availability of high-quality RNA structural data. & Data Quality Dependence: The method relies on accurate 3D RNA structural data, which may not always be available.**
>
> The definition of the RNA inverse design task dictates that **sequence generation must be based on existing three-dimensional structures** (such as the fixed backbone redesign task described in the paper). This limitation stems from the strong correlation between RNA function and its three-dimensional conformation. Current methods, including RBFN, require generating sequences conditioned on input structures, thus they cannot operate without the support of high-quality structural data. The current RNASolo dataset contains only about 18,955 RNA structures, as of 24-07-2025, which leads to a data scarcity issue in model training. Future improvements can be made in two directions:
>
> The first direction is, as the performance of RNA structure prediction tools improves, **predicted three-dimensional structures could supplement real data**. For example, similar to how ESM-IF[1] uses AlphaFold2-predicted protein structures for inverse folding training, RNA design could alleviate data bottlenecks by constructing virtual training sets based on predicted structures, thereby expanding sample diversity and improving performance.
>
> The second direction is to overcome data dependency, future work will explore **joint modeling of sequences and structures**, optimizing both simultaneously through multi-task learning, reducing reliance on single-sided data, much like existing works in protein design[2], aiming at the co-generation of RNA sequences and structures.
>
> [1] Hsu, Chloe, et al. "Learning inverse folding from millions of predicted structures." International conference on machine learning. PMLR, 2022.
>
> [2] Campbell, Andrew, et al. "Generative Flows on Discrete State-Spaces: Enabling Multimodal Flows with Applications to Protein Co-Design." International Conference on Machine Learning. PMLR, 2024.
>
> > **W3 & L3: Limited Wet-Lab Validation: There’s no real-world experimental validation of the method. & Lack of Wet-Lab Testing: Without experimental validation, the real-world applicability is uncertain.**
>
> We understand the reviewer's concern regarding wet lab validation. **Due to the current limitations of our laboratory conditions**, equipment resources, and experimental timelines, we are unable to independently complete functional validation experiments for RNA sequence design (such as in vitro folding validation, structural characterization, or functional activity testing) at this time. We are also very interested in conducting wet lab validations, and we commit to open-sourcing our code upon acceptance of the paper, **encouraging researchers worldwide to conduct validations in their own laboratories**. For example, we will provide standardized RNA sequence design templates to facilitate rapid experimentation by collaborating labs.
>
> > **Q1: Does the method work for more dynamic RNA structures?**
>
> Theoretically, RBFNs are capable of handling more dynamic RNA structures. The core idea is to **treat the dynamic conformations as a collection of multiple static states**. However, incorporating more conformations will increase the memory usage due to the expansion of the feature tensor dimensions.
>
> > **Q2 & L1: What are the computational costs for large-scale RNA design tasks? & Computational Cost: The method could be computationally expensive for large RNA molecules.**
>
> Thank you for your attention to the computational cost of RBFN. We re-ran the benchmarks on a GeForce RTX 3090 (24GB VRAM) and have made the test code publicly available. The experimental results show that **RBFN requires only 3.05 seconds and 1.84GB of VRAM for long RNA (4269nt)**, significantly outperforming gRNAde, which takes 66.28 seconds. These results demonstrate that RBFN can efficiently handle ultra-long RNA on consumer-grade GPUs, making it suitable for laboratory-level deployment.
>
> The following is how we add the test code for calculating time and memory usage:
>
> ```python
>
>     # Time & Memory Monitoring, Before Running the Model
>     torch.cuda.reset_peak_memory_stats(device)
>     time_start = time.time()
>
>     # Inference (50-step sampling, batch=1)
>     with torch.no_grad():
>         sequence = model.design(pdb_path, steps=50)
>
>     #After the operation is completed, get the time and GPU memory usage.
>     time_end = time.time()
>     peak_mem = torch.cuda.max_memory_allocated(device) / 1024**2  # MB
>     return time_end - time_start, peak_mem
> ```
>
> Here are the test results:
>
> | **Length (PDB ID)** | **gRNAde (time)** | **RBFN (time)** | **Memory (RBFN)** |
> | ------------------- | ----------------- | --------------- | ----------------- |
> | 25 (2A43)           | **0.3222s**           | 0.7063s    | 537.15 MB         |
> | 50 (6TF0)           | **0.6245s**           | 0.7026s    | 548.04 MB         |
> | 75 (2HOM)           | 0.9811s           | **0.6911s**     | 557.44 MB         |
> | 120 (3B4A)          | 1.5539s           | **0.7124s**     | 572.70 MB         |
> | 159 (6D8M)          | 2.0698s           | **0.7207s**     | 583.30 MB         |
> | 249 (9MEE)          | 3.2962s           | **0.7526s**     | 614.96 MB         |
> | **4269 (8TOC)**     | **66.2848s**      | **3.0469s**     | **1839.95 MB**    |

---

> ### Author Response · Authors · 2025-08-05
>
> Thank you for reviewing our paper. We noticed that you did not leave detailed comments, and we are eager to address any concerns you may have.
>
> If you have unresolved issues with the paper, we would greatly appreciate it if you could briefly outline them in a follow-up comment. We will carefully revise the manuscript to incorporate your feedback.
>
> Your input is invaluable to improving our work. Thank you again for your time and consideration!

---

> > ### Comment · Reviewer_LyRL · 2025-08-05
> > **Thanks for your response**
> >
> > Thanks for your response. I read the rebuttal and reviews from other reviewers carefully. Could you please provide the experimental results you mentioned in the response to reviewer qKvq? It would be important to validate the benefits of using BFN over other methods.

---

> > > ### Author Response · Authors · 2025-08-06
> > >
> > > Thank you for your reply. We have supplemented the comparison methods. Regarding the selection of the generative model, we will conduct it in future work. This may be a general problem with the generative model. The motivation for us to use BFN is its performance on discrete data [1][2][3].
> > >
> > > [1] Song, Yuxuan, et al. "Unified generative modeling of 3d molecules with bayesian flow networks." The Twelfth International Conference on Learning Representations. 2024.
> > >
> > > [2] Qu, Yanru, et al. "MolCRAFT: Structure-Based Drug Design in Continuous Parameter Space." Forty-first International Conference on Machine Learning. 2024.
> > >
> > > [3] Gong, Jingjing, et al. "Steering Protein Family Design through Profile Bayesian Flow." ICLR. 2025.

---

> > > > ### Comment · Reviewer_LyRL · 2025-08-06
> > > >
> > > > I noticed that in your response to another reviewer, you mentioned that the additional experiments are “currently implementing a comparative experiment.”
> > > > However, in your reply to my question about the results, you indicated that the experiments will be conducted in the future.
> > > > To help us fully assess the manuscript, could you please clarify the current status of these experiments?
> > > > If any preliminary results are already available, it would be helpful to include them. If the experiments have not yet started, please let us know the reason and the expected timeline, so we can better understand their relevance to the conclusions.

---

> > > > > ### Author Response · Authors · 2025-08-06
> > > > >
> > > > > Thank you for your reply. We are trying to use diffusion for comparison, and we will try our best to submit the comparison of relevant results (results of diffusion vs BFN on this task) before the deadline. Regarding the selection of generative models for general tasks, we think it is a research direction worthy of exploration.

---

> > > > > ### Author Response · Authors · 2025-08-08
> > > > >
> > > > > We implemented a version of RBFN(Diffusion) using the method of implementing the discrete diffusion model in [1]. We used a 1000-step training deployment and generated through the sampling method of the standard diffusion model. The experimental results are as follows:
> > > > >
> > > > > | PDB ID | Description                                | ViennaRNA | FARNA | RDesign | Rosetta | gRNAde     | RBFN       | RBFN(Diffusion) |
> > > > > |--------|--------------------------------------------|-----------|-------|---------|---------|------------|------------|-----------------|
> > > > > | 1CSL   | RRE high affinity site                     | 0.25      | 0.20  | 0.4455  | 0.44    | **0.5719** | 0.4832     | 0.4583          |
> > > > > | 1ET4   | Vitamin B12 binding RNA aptamer            | 0.25      | 0.34  | 0.3929  | 0.44    | 0.6250     | 0.4482     | **0.6667**      |
> > > > > | 1F27   | Biotin-binding RNA pseudoknot              | 0.30      | 0.36  | 0.3013  | 0.37    | 0.3437     | **0.5208** | 0.4583          |
> > > > > | 1L2X   | Viral RNA pseudoknot                       | 0.24      | 0.45  | 0.3727  | 0.48    | 0.4721     | **0.5833** | 0.5417          |
> > > > > | 1LNT   | RNA internal loop of SRP                   | 0.33      | 0.27  | 0.5556  | 0.53    | 0.5843     | **0.6375** | 0.4688          |
> > > > > | 1Q9A   | Sarcin/ricin domain from E.coli 23S rRNA   | 0.27      | 0.40  | 0.4417  | 0.41    | 0.5044     | **0.8588** | 0.4861          |
> > > > > | 4FE5   | Guanine riboswitch aptamer                 | 0.29      | 0.28  | 0.4112  | 0.36    | 0.5300     | 0.5326     | **0.5612**      |
> > > > > | 1X9C   | All-RNA hairpin ribozyme                   | 0.26      | 0.31  | 0.3967  | 0.50    | 0.5000     | **0.5187** | 0.4337          |
> > > > > | 1XPE   | HIV-1 B RNA dimerization initiation site   | 0.27      | 0.24  | 0.3834  | 0.40    | 0.7037     | 0.6522     | **0.7917**      |
> > > > > | 2GCS   | Pre-cleavage state of glmS ribozyme        | 0.25      | 0.26  | 0.4518  | 0.44    | **0.5078** | 0.4990     | 0.4167          |
> > > > > | 2GDI   | Thiamine pyrophosphate-specific riboswitch | 0.25      | 0.38  | 0.3523  | 0.48    | **0.6500** | 0.6042     | 0.6146          |
> > > > > | 2OEU   | Junctionless hairpin ribozyme              | 0.23      | 0.30  | 0.5000  | 0.37    | **0.9519** | 0.5580     | 0.7083          |
> > > > > | 2R8S   | Tetrahymena ribozyme P4-P6 domain          | 0.27      | 0.36  | 0.5641  | 0.53    | 0.5689     | **0.7172** | 0.5169          |
> > > > > | 354D   | Loop E from E. coli 5S rRNA                | 0.28      | 0.35  | 0.4458  | 0.55    | 0.4410     | **0.8031** | 0.5931          |
> > > > > |        | Overall recovery:                          | 0.27      | 0.32  | 0.4296  | 0.45    | 0.5682     | **0.6012** | 0.5512          |
> > > > >
> > > > >
> > > > > The experimental results show that RBFN (Diffusion) exhibits relatively good performance on some results, but overall, its performance is inferior to the original RBFN. To some extent, this reflects that BFN performs better than Diffusion in the RNA inverse design task. The method of generating discrete data through a continuous parameter space is also proven in [2][3][4]. However, this does not mean that Diffusion with more training techniques integrated will be weaker than BFN. We would like to emphasize this point.
> > > > >
> > > > > [1] Hoogeboom, Emiel, et al. "Equivariant diffusion for molecule generation in 3d." International conference on machine learning. PMLR, 2022.
> > > > >
> > > > > [2] Song, Yuxuan, et al. "Unified generative modeling of 3d molecules with bayesian flow networks." The Twelfth International Conference on Learning Representations. 2024.
> > > > >
> > > > > [3] Qu, Yanru, et al. "MolCRAFT: Structure-Based Drug Design in Continuous Parameter Space." Forty-first International Conference on Machine Learning. 2024.
> > > > >
> > > > > [4] Gong, Jingjing, et al. "Steering Protein Family Design through Profile Bayesian Flow." ICLR. 2025.

---

> > > > > > ### Comment · Reviewer_LyRL · 2025-08-08
> > > > > >
> > > > > > Thanks for your response. I stand by my original assessment and encourage the authors to include these findings from the discussion in the next version.

---

> > > > > > > ### Author Response · Authors · 2025-08-08
> > > > > > >
> > > > > > > Thank you for your reply. We believe these will make our work more complete and better presented to readers.

---

### Official Review · Reviewer_DZoU · 2025-06-22

**Clarity:** 2
**Significance:** 2
**Originality:** 3
**Rating:** 3
**Confidence:** 3

**Summary:**

This paper introduces RBFN, a novel generative model for the RNA inverse design problem, which aims to generate valid nucleic acid sequences for a given 3D RNA structure. It combines a geometric-algebra-enhanced graph neural network with a Bayesian flow network (BFN) and addresses two key challenges: the "one-to-many" nature of the problem and the conformational flexibility of RNA. The work provides clear theories and promising experimental results.

**Questions:**

1. What is the metric used in Table 1?

2. In Table 2, does AR refer to autoregression? The RBFN is only compared with the baseline gRNAde with AR. Are there additional comparisons with other models for this task, such as diffusion models or VAEs? Why not compare RBFN with the methods from Table 1?

3. Is there any demonstration that the geometric algebra module provides geometric inductive bias? Why do the multiple vectors 0/1/2/3-... improve expressivity? Or the performance gain may come from the increasing parameters.

**Ethical Concerns:**

["NO or VERY MINOR ethics concerns only"]

**Final Justification:**

I have reviewed the authors' response. Overall, it addresses the concerns reasonably well, but I find the motivation still unconvincing, largely due to its presentation. While the bullet points in the rebuttal help clarify some points, the authors did not clearly revise specific lines or paragraphs to improve readability, aside from a few high-level changes. As a result, it is difficult to assess the effectiveness of the revision, so I am keeping my score as borderline.

**Limitations:**

yes

**Quality:**

2

**Strengths And Weaknesses:**

Strengths:

1. The paper introduces geometric algebra to encode RNA 3D backbone geometry, yielding richer structural representations than traditional coordinate or hand-crafted features.

2.  The work has effective ablation studies, removing either the geometric algebra module or the custom sampling schedule degrades performance markedly. The reports with mean and standard deviations add confidence that gains aren’t due to random chance.

3.  The results show gains in diversity metrics (UCR/USR). They support the major hypothesis that a generative flow-based model is better suited for exploring the vast sequence space of RNA.

Weaknesses:

The paper should explain the motivation for using geometric algebra and a Bayesian Flow Network. It currently does not describe in detail the limitations of the existing RNA modeling methods used as baselines, which components those approaches lack, and why geometric algebra together with a Bayesian Flow Network can address these gaps.

---

> ### Author Rebuttal · Authors · 2025-07-29
>
> Dear reviewer DZoU,
>
> We express our gratitude for your diligent review and the valuable efforts you have made to assist us in enhancing the manuscript.
>
> > **W: Motivation for using geometric algebra and a Bayesian Flow Network**
>
> We sincerely thank the reviewers for their in-depth questions regarding the motivation behind our method and the limitations of existing RNA modeling approaches. Below, we will systematically elaborate on four aspects: **the shortcomings of current methods**, **the rationale for introducing GA**, **the design logic of BFN**, and the synergistic mechanism by which both **address key issues**. We will also illustrate the innovativeness of RBFN with experimental results. The relevant content will be updated in the subsequent version of the paper.
>
> **1. Core Limitations of Existing RNA Modeling Methods**
>
> The current mainstream RNA design methods (such as Rosetta) face the following critical bottlenecks:
>
> **(a) Shallow 3D Structure Modeling**
>
> Existing tools over-rely on secondary structure predictions (e.g., ViennaRNA), neglecting tertiary interactions. This necessitates error-prone manual feature engineering (e.g., defining distance/torsion constraints in Rosetta), leading to poor generalization to complex conformations and noise sensitivity.
>
> **(b) Inefficiency in Discrete Sequence Generation**
>
> Autoregressive decoding (e.g., gRNAde) suffers from cumulative errors in long sequences (>500nt), where early prediction errors cascade through attention mechanisms. Additionally, insufficient distribution modeling treats sequence generation as discrete optimization, lacking probabilistic sampling of diverse, stable RNAs, resulting in low diversity/compatibility.
>
> **(c) Absence of Multi-State Modeling**
>
> Existing tools (e.g., RDesign) mostly focus on designing single states with fixed backbones, whereas biologically active RNAs (e.g., riboswitches) often need to dynamically switch between multiple conformations.
>
> **2. Motivation and Advantages of Introducing GA**
>
> Geometric algebra (GA) intrinsically encodes geometric inductive biases through its mathematical structure, directly addressing the 3D modeling limitations in existing methods. (See Q3 response for details.)
>
> **3. Design Logic and Advantages of BFN**
>
> We adopted Bayesian Flow Networks (BFN) based on its demonstrated success in handling discrete tasks such as molecular generation [1], drug design in continuous parameter space [2], and protein sequence design [3]. BFN addresses the efficiency and diversity issues in discrete sequence generation by evolving distributions in a continuous parameter space:
>
> **(a) Mechanism of Continuous Time Distribution Evolution**
>
> BFN models sequence generation as the evolution of a probability distribution over time ( $t \in [0,1]$ ), directly sampling complete sequences via continuous gradient optimization, thus eliminating cumulative errors in autoregressive decoding.
>
> **(b) Low-Variance Sampling and Small Alphabet Adaptation**
>
> BFN directly models the categorical distribution of the four bases, minimizing the KL divergence between the model distribution and the true data distribution. By leveraging continuous gradients, it enhances sampling efficiency, resulting in a higher sequence recovery rate (60.12%) for RBFN compared to gRNAde (56.82%).
>
> [1] Song, Yuxuan, et al. "Unified generative modeling of 3d molecules with bayesian flow networks." The Twelfth International Conference on Learning Representations. 2024.
>
> [2] Qu, Yanru, et al. "MolCRAFT: Structure-Based Drug Design in Continuous Parameter Space." Forty-first International Conference on Machine Learning. 2024.
>
> [3] Gong, Jingjing, et al. "Steering Protein Family Design through Profile Bayesian Flow." ICLR. 2025.
>
> **4. Synergistic Mechanism of Geometric Algebra and BFN**
>
> The 3D structural features encoded by the GA module (e.g., bivector plane relationships) **serve as the prior distribution** for BFN, guiding sequence generation to ensure spatial compatibility. During BFN’s evolution process, the GA module continuously validates structural feasibility and corrects deviations via gradient backpropagation, enabling dynamic feedback.
>
> In the revised manuscript, we will strengthen the explanation of the method's motivation at Introduction, clearly outlining the limitations of existing methods and the targeted improvements of RBFN in the introduction and related work sections. We believe that RBFN, through the deep integration of Geometric Algebra and Bayesian Flow Networks, provides a new methodological framework for the field of RNA design.
>
> > **Q1: Metric in Table1**
>
> Native Sequence Recover, we will update the version and place it in the table caption. Thank you for pointing out this issue.
>
> > **Q2: Table2's explain and compare methods**
>
>
> Yes, AR (Autoregressive) in Table 2 refers to Autoregressive Modeling.
>
> In the original submission, RBFN was primarily compared with gRNAde (the current SOTA model) because its publicly available code and evaluation metrics facilitated direct comparison. However, during the Rebuttal phase, we further supplemented the following experiments:
>
> - RhoDesign: An RNA design method based on 3D structure.
>
> - RiboDiffusion: An RNA design method based on a diffusion model, which generates sequences by progressively reversing noise.
>
> Due to space limitations, the results of the additional comparative experiments can be found in the Rebuttal for qKvq.
>
> The methods in Table 1 are mainly based on secondary structure for generating RNA sequences and perform poorly in 3D structure. Therefore, we primarily compared with the SOTA method, gRNAde. Additionally, to demonstrate the improvement of RBFN in 3D structure metrics (such as scTM-score), we retained the comparison with ViennaRNA (a classic 2D tool).
>
> > **Q3: geometric inductive bias and improve expressivity**
>
> We are grateful for the reviewer's in-depth attention to the GA. We provide a targeted response to this issue:
>
> **1. Does the Geometric Algebra module provide geometric inductive bias?**
>
>
> Yes, GA intrinsically embeds powerful geometric inductive bias into the model, significantly enhancing its capability to represent complex RNA three-dimensional structures. This advantage stems directly from GA's mathematical structure, which encodes physical-world geometric relationships through:
> **1) Hierarchical multivectors** (scalars, vectors, bivectors, trivectors/pseudoscalars) and
> **2) Fundamental geometric operations** (outer product, inner product, rotors).
> Critically, each multivector level captures distinct geometric semantics:
> - **Vectors (1-vectors)** represent directional displacement (e.g., distances between atoms like C4' and P);
> - **Bivectors (2-vectors)** characterize planar orientation (e.g., base pair relationships);
> - **Trivectors/pseudoscalars (3-vectors)** model volumetric properties and global conformational changes.
>
> This hierarchical representation enables the model to **directly learn** essential spatial patterns—distances, angles, and torsion angles—**without relying on error-prone manual feature engineering**. The critical role of GA's inductive bias is unequivocally demonstrated in ablation studies: removing GA features ("w/o GA") caused a significant performance drop, reducing sequence recovery rate from 60.1% to 57.0%. These results confirm that GA's geometric priors are fundamental to the model's enhanced ability to capture and exploit RNA structural information.
>
>
> **2. How do multivectors (0/1/2/3...) enhance expressive power?**
>
>
>  **The hierarchical geometric semantics and compositional capabilities of multivectors fundamentally enhance the expressive power of RNA structure models.** Geometric Algebra achieves this through its graded multivector structure, where each level captures distinct physical properties: _scalars_ encode global attributes like energy states; _vectors_ represent directional features (e.g., atomic displacements like C3'→P distances); _bivectors_ define planar orientations critical for base-pairing geometries; and _trivectors/pseudoscalars_ characterize volumetric relationships governing RNA tertiary conformations. These geometric primitives combine compositionally through operations like the geometric product (uv=u⋅v+u∧v), enabling simultaneous modeling of symmetric (scalar) and antisymmetric (bivector) interactions – a capability absent in traditional RNA design frameworks. Further amplifying this expressivity are GA's **inherent operational advantages**:
>
>  - The **outer product** (u∧v) constructs higher-dimensional subspaces (e.g., generating a bivector plane from two vectors), capturing nonlinear geometric relationships essential for complex motifs;
>  - **Rotors** implement efficient spatial transformations (R=e−θB/2) without Euler/matrix decomposition, streamlining conformational modeling;
>  - **Parameter efficiency** emerges from GA's compact representation – a single bivector in G3,0,0 space can encode rotational and planar data that would require multiple disconnected parameters in classical GNNs. This unified mathematical framework thus provides both richer structural representation and greater computational economy for RNA design tasks.
>
>
>
> **3. Is the performance improvement due to an increase in the number of parameters?**
>
> The performance improvement is mainly attributed to the structural advantages of Geometric Algebra, rather than a mere increase in the number of parameters. gRNAde (with 1,104,311 parameters) achieved only a 56.82% sequence recovery rate in the single-state task, while RBFN (with 1,106,977 parameters) reached 60.12%, indicating that the structural design of Geometric Algebra is more efficient.

---

> > ### Comment · Reviewer_DZoU · 2025-08-04
> >
> > Thank you for the authors' detailed response. The motivation is now clearer to me, and there is no further concern except for presentation. The use of bullet points in the rebuttal is not an effective way to clarify the motivation in the paper. A substantial revision is needed to present the motivation more clearly in the main text, particularly regarding Geometric Algebra and Bayesian Flow Networks, and to make it more accessible to readers from the broader machine learning community who are interested in RNA modeling.

---

> ### Author Response · Authors · 2025-08-04
>
> Thank you for your detailed feedback and for acknowledging our clarifications in the Rebuttal regarding the motivation section, as well as for resolving your concerns except for presentation. We clarify that the bullet-point format was adopted in the Rebuttal due to word limit constraints to efficiently address your concerns, but this should not replace the fluid narrative required in the main text.
>
> As you emphasized, the current manuscript fails to sufficiently contextualize **Geometric Algebra (GA)** and **Bayesian Flow Networks (BFNs)** for the broader machine learning community, particularly researchers focused on RNA modeling. To address this, we will implement the following precise revisions to ensure content accessibility:
>
> - **Strengthen motivation and problem-driven logic in the Introduction**:
>
>   - **Systematically discuss existing limitations**: Explicitly highlight three key shortcomings of current RNA modeling approaches:
>
>     *Shallow 3D Structure Modeling*, *Inefficiency in Discrete Sequence Generation*, and *Absence of Multi-State Modeling*.
>
>   - **Elaborate motivation in detail**: Introduce the necessity of GA and explain how it resolves these issues. Illustrate BFNs’ advantages using noise sensitivity in RNA sequence generation. Avoid introducing abstract theories first, ensuring ML readers intuitively understand *"why these methods are needed"*. **We have added citations to existing methods to validate effectiveness**.
>
> - **Add a new section in the appendix titled "RNA Structural Characteristics and Modeling Challenges"**:
>
>   - Provide background knowledge on RNA structure to help the broader machine learning community understand the context.
>
>   - Clarify the rationale and effectiveness of GA by explaining its advantages and logical connections to RNA modeling.
>
> These revisions will establish a **dual-layer support system**: The Introduction will deliver a problem driven motivation narrative (accessible to all ML researchers), while the appendix will supplement domain background (lowering the RNA knowledge barrier). Together, they will ensure a broader audience can smoothly grasp the innovative value of GA/BFNs.
>
> We sincerely appreciate your invaluable suggestions, which have significantly improved the manuscript's presentation and enhanced its accessibility for readers from the broader machine learning community.

---

### Official Review · Reviewer_jqC4 · 2025-06-25

**Clarity:** 1
**Significance:** 2
**Originality:** 2
**Rating:** 3
**Confidence:** 3

**Summary:**

The paper introduces a novel model for RNA inverse design, i.e. the generation of RNA sequences given one (or several) 3D backbone structures. The model is based on the generative framework of Bayesian Flow networks, which allow to learn the distribution of discrete RNA sequences. The network architecture uses layers from recent graph neural networks based on Clifford algebra to enhance the geometric expressiveness of the model.  The paper further employs a new distribution for the sampling of time-steps which emphasizes times near t=0. The model achieves competitive result on various metrics in single- and multi-state RNA design benchmarks.

**Questions:**

- I think it would be good to include a plot of the beta distribution used for the time-step distribution for better intuition on the shape of the distribution
- On page 4 it is indicated that for the neighborhood of a node the 32 nearest neighbors are used. I wonder why a fixed number of neighbors was chosen instead of defining a cutoff radius?
- How long does inference take for models with different numbers of steps and compared to the baseline models (in particular gRNAde)

**Ethical Concerns:**

["NO or VERY MINOR ethics concerns only"]

**Final Justification:**

I had many concerns about the presentation of the original submission, which had a lot of missing information and undefined expressions which made it very hard to follow the logic of the paper. During rebuttal the authors clarified many of these points. I remain hesitant to vote for acceptance of the work, since in my opinion a resubmussion would allow for a more accurate judgement of the presentation quality of the revised paper. Due to the improvements made during rebuttal I have raised my score to 3.

**Limitations:**

Yes

**Quality:**

2

**Strengths And Weaknesses:**

- Strengths:
    - The paper has a clear target in improving RNA inverse design and overcoming common problems in this task like the existence of several conformers and target sequences for a given 3D structures. The proposed methods are sensible and are proven to be helpful in the ablation studies.
    - The paper does provide errors obtained from runs with three different random seed for most conducted experiments
- Weaknesses:
    - It is very hard to follow the formal presentation of methods in the paper. The main problem here is that a large number of symbols and expressions are not properly defined anywhere which makes the framework only understandable with prior knowledge in both Geometric algebra and Bayesian flow networks. Below is an incomplete list of examples:
        - There is no clear notation for scalars, vectors and higher grade elements of Clifford algebra. Sometimes vectors are written in bold sometimes not. Sometimes they are written with an arrow on top sometimes they are not.
        - On page 3 , G is introduced as the 3D backbone structure, but the components of the tuple are not defined until section 3.1 and they also do not reference the backbone coordinates in the previous sentence.
        - The notation for a Clifford algebra G_{p,q,r} and the parameters p,q,r are mentioned but never properly defined
        - In the first paragraph of page 5 the index k is reused multiple times and sometimes wrongly instead of the correct index n (for the number of graphs). Other indices like f and f’ are never defined
        - In line 165 the initialization of geometric algebra features is unclear, the symbol q was also never defined before
        - …
    - It is never explicitly stated which Clifford algebra is used within the model. I assume its the Euclidean geometric algebra since this is what the reference architecture uses. In this case the interpretation of elements as points, lines, planes and solid structures is wrong. Points, lines and planes would be appropriate interpretations for projective geometric algebra, whereas the geometric interpretation of Elements of the Euclidean geometric algebra corresponds to scalars, vectors, directed area elements (pseudovectors) and directed volume elements (pseudoscalars). A discussion for why this algebra was chosen over other geometric algebras which have also found use in recent deep learning architectures is missing
    - Overall the results of RBFN are only marginally better than the baselines. Given that inference time is probably higher due to the iterative procedure, the use cases for the model are limited.
    - There is the related work [1] which is not cited, but seems to achieve similar results on the single-state benchmark dataset

[1] Manzourolajdad, A., & Mohebbi, M. (2025). Secondary-Structure-Informed RNA Inverse Design via Relational Graph Neural Networks. Non-coding RNA, 11(2), 18.

---

> ### Author Rebuttal · Authors · 2025-07-30
>
> Dear reviewer jqC4,
>
> We express our gratitude for your diligent review and the valuable efforts you have made to assist us in enhancing the manuscript. We denote geometric algebra by GA.
>
> > **W1: Question about formal presentation**
>
> Thank you to the reviewer for pointing out this issue. We have revised the paper and will update it in the updated manuscript. Due to the word limit, we provide a brief summary of the revisions:
>
> - For **section 2.1 Problem Definition**, to avoid content confusion, we removed the introduction of G and revised the description to re-explain nucleotides so that readers can understand the structure of nucleotides and the content of backbone atoms. The specific symbol content has been moved to section 3.1.
>
> - For **section 2.2 GA**, we have rephrased the language and added explanations for the symbols, for example:
>
> '''
>
>  $\mathbb{G}_{p,q,r}$ denotes a GA over a real vector space with $p$ basis vectors of positive signature ($e_i^2 = +1$), $q$ of negative signature ($e_j^2 = -1$), and $r$ of zero signature ($e_k^2 = 0$), satisfying $p + q + r = n$ for an $n$-dimensional space.
>
> '''
>
> - **For section 2.3  Bayesian Flow Networks**, we have reorganized the relevant formulas using the symbols and expressions from diffusion models to make it easier for readers to understand. For example:
>
> '''
>
> BFNs introduce a continuous-time parameter $t\in[0,1]$ and define two key distributions: (1) the \textit{sender distribution} $q_t(\mathbf{x}_t)$ representing the data distribution at time $t$, and (2) the \textit{receiver distribution} $q_t(\mathbf{x}_t|\mathbf{x}_0)$ describing the forward process from the original data $\mathbf{x}_0$ to the corrupted data $\mathbf{x}_t$. The core of BFNs is the Bayesian inversion mechanism, where the reverse process is given by:
>
> $q_t(\mathbf{x}_0|\mathbf{x}_t) = \frac{q_t(\mathbf{x}_t|\mathbf{x}_0)q_0(\mathbf{x}_0)}{q_t(\mathbf{x}_t)}$
>
> '''
>
> - **For section 3.1 GA GNN for RNA Three-Dimensional Structure Modeling**, we have provided a more detailed explanation of the symbols and have re-unified some individual symbols. Additionally, we have added symbolic identification for coordinates related to atoms, for example:
>
> '''
>
> For an RNA structure with $m$ conformational states and $n$ nucleotides, we represent it as $m$ graphs $\{ \mathcal{G}^{(1)}, \dots, \mathcal{G}^{(m)}\}$.
>
> ...
>
> $\mathfrak{q}(\mathbf{v})$ is obtained based on the three-dimensional coordinate relationships. For a multivector $\mathbf{v} = s + \vec{v} + \mathbf{b} + t\mathbf{I}$ where $s$ is scalar, $\vec{v}$ vector, $\mathbf{b}$ bivector, and $t\mathbf{I}$ pseudoscalar components, the quadratic form is $\mathfrak{q}(\mathbf{v}) = s^2 + \|\vec{v}\|^2 - \|\mathbf{b}\|^2 - t^2$ in $\mathcal{G}_{3,0,0}$.
>
> '''
>
> - **For section 3.2 on Bayesian Flow Networks for RNA Generation**, we have concretized the relevant symbols and updated the descriptions to make them more focused on the task of RNA design, thereby making it easier for readers to understand. For example:
>
> '''
>
> Before training, we define a \textbf{uniform prior} over the 4 RNA nucleotides: $p_0(\mathbf{y}) = \text{Categorical}(\mathbf{y}|\pi_0)$ with $\pi_0 = [1/4, 1/4, 1/4, 1/4]^\top$.
>
> ...
>
> '''
>
> **We hope that these detailed revisions and enhancements will make the methods section easier for readers to understand, thereby contributing to scientific research.**
>
> > **W2: Question about Clifford algebra**
>
> Thank you for your professional feedback on the use of Clifford algebra in our paper. We will make the necessary revisions in the updated manuscript.
>
> You accurately identified a key conceptual confusion in our paper: in Figure 2, we incorrectly used the interpretative framework of projective GA, misinterpreting 0-vectors, 1-vectors, 2-vectors, and 3-vectors as "points, lines, planes, and solid structures," respectively.
>
> As you correctly pointed out, in standard 3D Euclidean GA ($\mathcal{G_{3,0,0}}$):
>
> - scalars represent quantities without direction
>
> - vectors represent directed line segments
>
> - bivectors represent directed planes/area elements
>
> - trivectors/pseudoscalars represent directed volumes
>
> We acknowledge that this was an oversight in our presentation, stemming from a confusion between different variants of GA. While we used Euclidean GA for calculations in our model design, we inadvertently adopted the common interpretative framework of projective geometry from computer graphics when explaining its biological significance. The intuitive interpretation of nucleotide positions in RNA molecular structures led us to erroneously map Euclidean GA elements onto the projective geometric interpretation.
>
> We will make the following significant revisions in the updated manuscript:
>
> 1. Fully correct the interpretation in Figure 2, clearly distinguishing the proper interpretative framework of Euclidean GA.
>
> 2. Add a dedicated subsection (Section D.3) to discuss why we chose $\mathcal{G_{3,0,0}}$ over other GAs (such as projective GA $\mathcal{G_{3,0,1}}$ or conformal GA  $\mathcal{G_{4,1,0}}$):
>
> - **Spatial Representation**: RNA structures reside in standard 3D Euclidean space, where $\mathcal{G}_{3,0,0,}$ directly models spatial relationships without extra dimensions. RNA design focuses on nucleotide relative positions, orientations, and local geometry naturally captured by vectors and bivectors in Euclidean GA.
> - **Task Alignment**: While projective GA (PGA) excels at representing points/planes, RNA design prioritizes vector relationships and local geometry over absolute positions. $\mathcal{G}_{3,0,0}$ balances mathematical simplicity with sufficient expressiveness for key RNA structural features.
> - **Computational Efficiency**: $\mathcal{G}_{3,0,0}$ (8 dimensions) offers lower complexity than PGA (16D) or conformal GA (32D). This efficiency is critical for RNA inverse design involving long sequences (thousands of nucleotides).
>
>
> 3. Re-examine all sections of the paper that involve geometric interpretations to ensure conceptual consistency.
>
> We appreciate your identifying this critical issue. Your feedback helped us correct conceptual errors and deepened our understanding of the theoretical foundations of GA in RNA structure modeling. Your professional insights significantly enhanced the rigor and scientific value of our work.
>
> > **W3&Q3: Performance & inference time of RBFN**
>
> Thank you for your attention to performance and the computational cost of RBFN.
>
> While RBFN shows modest but meaningful improvements over gRNAde, the gains align with RNA’s structural complexity:
>
> - 5.8% higher recovery (0.6012 vs. 0.5682),
> - 17.0% lower scRMSD error (10.83 vs. 13.04).
>
> Given RNA’s extreme flexibility and shallow energy landscapes—where sub-angstrom precision dictates function—even incremental gains in structural accuracy (e.g., 2.21 Å scRMSD reduction) directly impact biological interpretability.
>
>
>
> **For the computational cost**, we re-ran the benchmarks on a GeForce RTX 3090. The experimental results show that RBFN requires only 3.05 seconds for long RNA (4269nt), **21.7 times faster than** gRNAde(AR), which takes 66.28 seconds. These results demonstrate that RBFN can efficiently handle ultra-long RNA on consumer-grade GPUs, making it suitable for laboratory-level deployment.
>
>
> Here are the test results:
>
> | **Length (PDB ID)** | **gRNAde (time)** | **RBFN (time)** |
> | ------------------- | ----------------- | --------------- |
> | 25 (2A43)           | **0.3222s**           | 0.7063s     |
> | 50 (6TF0)           | **0.6245s**          | 0.7026s     |
> | 75 (2HOM)           | 0.9811s           | **0.6911s**     |
> | 120 (3B4A)          | 1.5539s           | **0.7124s**     |
> | 159 (6D8M)          | 2.0698s           | **0.7207s**     |
> | 249 (9MEE)          | 3.2962s           | **0.7526s**     |
> | **4269 (8TOC)**     | **66.2848s**      | **3.0469s**     |
>
> > **W4&Q1: related work & a plot of the beta distribution**
>
> We will supplement the **revision with a citation and discussion** of this literature to more comprehively present the current progress and challenges in the research, and we will also **add a plot of the beta distribution**. Using secondary structure is a great idea, which has also given us some inspiration. In our future work, we will further explore how to utilize this information.
>
> > **Q2: Using 32 nearest neighbors instead of a fixed truncation radius**
>
>
>
> Thank you for this insightful question. We use fixed neighbor counts (32) instead of distance cutoffs to address fundamental challenges in RNA modeling:
>
> 1.  **Solving Density Heterogeneity:**
>     RNA regions show extreme density variations:
>     - **Stems:** Tight packing (P-P ~5.9 Å → 6-10 neighbors at 5Å cutoff)
>     - **Loops:** Loose packing (P-P 7-8 Å → 4-7 neighbors)
>     - **Pseudoknots:** Ultra-dense (→15-20+ neighbors)
>     Fixed cutoffs cause **unstable neighbor counts**, risking redundancy in dense areas and missed interactions in sparse regions.
>
> 2.  **Mathematical Necessity for GA-GNN:**
>     - **32 radial basis functions (RBFs)** encode distances.
>     - **Matching neighbor count (32)** ensures each RBF interval is sufficiently sampled.
>     - Critical for stable **geometric product calculations** and **multivector normalization**.
>
> 3.  **Biophysical Justification:**
>     - Core functional interactions occur within **~15-20 sequence neighbors** (local motif size).
>     - **32 neighbors** intentionally **captures essential long-range tertiary contacts** while aligning with RBF dimensions.
>
>
> In summary, the strategy of using a fixed number of neighbors allows our model to **robustly handle the density variability of RNA structures** while maintaining the mathematical rigor of GA representations.

---

> > ### Comment · Reviewer_jqC4 · 2025-08-01
> >
> > I thank the authors for their extensive responses and additional experiments. However, I remain unconvinced by certain aspects of the work, which I detail below.
> >
> > In general I appreciate the measures that were taken to enhance the clarity of the text. Unfortunately, it is hard to judge the impact on the comprehensibility of the paper as a whole, due to the nature of the rebuttal format.
> >
> > I also still think that the discussion of the GA based architecture is more confusing than helpful, particularly for readers unfamiliar with GA. The paper directly applies the architecture described in [27] without introducing signigicant architectural modifications. Together with the fact that the maths in section 3.1 is also largely copied from [27] I believe it would be more appropriate to only provide an intuitive overview in the main text and move the details to the Appendix.
> >
> > In general, I find the use of BFNs much more convincingly motivated than the inclusion of GA. As Euclidean geometric algebra is a quotient algebra of tensor algebra, it does not inherently provide a computational advantage over methods based on equivariant tensor product layers. I would thus argue that the actual architectural choice may not be crucial for the performance of the method, and other state-of-the-art geometric GNNs would perform equally well. The conducted ablation study “w/o GA” is also not too insightful in that regard, since removing the GA features deprives the network of all equivariant features, which is of course expected to degrade its performance without offering interesting insights into the performance of GA over other equivariant features. My suggestion would thus be to discuss GA in the context of the well tuned architecture introduced in [27] rather than an integral part of the pipeline.
> >
> > Overall, I acknowledge the potential of the paper, especially the use of BFNs for RNA inverse design, but I currently believe that a more substantial revision and resubmission would be most appropriate, which would then also allow to properly reevaluate the comprehensibility of the paper as a whole.

---

> ### Author Response · Authors · 2025-08-02
>
> We thank the reviewer for their detailed feedback on our manuscript. We appreciate the time and effort taken to evaluate our work and provide constructive suggestions. Here, we systematically elaborate on the core advantages of GA as a structural feature encoder for RNA, its alignment with task-specific characteristics, and clarify the research positioning of this work.
>
> The technical necessity of GA in RNA modeling stems from the intrinsic geometric properties of RNA structures. RNA molecules exhibit hierarchical geometric constraints: helical symmetry of the phosphate backbone, planar rigidity of base pairs, continuous distributions of torsion angles, and topological folding in tertiary structures. Traditional Euclidean representations require complex feature engineering or redundant parameterization to approximate these constraints, whereas **GA’s mathematical structure enables models to effectively learn these geometric patterns**:
>
> 1. **Learning rotation invariance**: GA’s rotor representation can be efficiently learned by models to automatically encode rigid rotations of base pairs, avoiding information loss during coordinate transformations.
>
> 2. **Explicit modeling of structural constraints**: Geometric features generated by the wedge product (e.g., base plane normal vectors, hydrogen bond directions) can be directly optimized by models, enhancing learning capability for high-dimensional geometric relationships.
>
> 3. **Unified learning of multiscale features**: GA’s multivector hierarchy (scalar-vector-bivector) allows models to simultaneously learn atomic coordinates and secondary/tertiary topological information (e.g., stem-loops, pseudoknots).
>
> These properties establish GA as a task-adaptive learning tool for RNA structure learning. The core contribution of using GA is the first validation of the learnability of GA features in RNA sequence design tasks, the experimental results confirm this. Additionally, the reviewer commented that "other state-of-the-art geometric GNNs would perform equally well." Is there any basis for this statement? Could you provide some references to verify it?
>
> In response to the reviewer’s comments, we have further revised the paper structure to enhance readability, with an emphasis on the representation of GA in RNA modeling. For example:
>
> '''
>
> For an RNA structure with $m$ conformations and $n$ nodes, consider this structure as multiple graphs, i.e., $\{ \mathcal{G}^{(1)}, \dots, \mathcal{G}^{(m)}\}$. For each conformation graph $\mathcal{G}^{(c)} = ( \mathbf{A}^{(c)}, \mathbf{S}^{(c)}, \vec{\mathbf{X}}^{(c)} )$ where $c \in \{1,\dots,m\}$, it contains scalar features, namely the index and sequence representation of nucleotides in RNA $\mathbf{S}\in\mathbb{R}^{n\times m \times f}$, and vector features $\vec{\mathbf{X}} \in \mathbb{R}^{n \times m \times f' \times 3}$: (a) the forward and backward unit vectors along the backbone from the 5' end to the 3' end, ($\vec{\mathbf{x}}\_{i +1}-\vec{\mathbf{x}}\_{i}$ and $\vec{\mathbf{x}}\_{i}-\vec{\mathbf{x}}\_{i-1}$); and (b) the unit vectors, distances, angles, and torsions from each C4' to the corresponding P and N1/N9.
> The edge features $\{ \mathbf{A}^{(1)}, \dots, \mathbf{A}^{(m)} \}$ from node $j$ to $i$ are initialized as follows: (a) the unit vector from the source node to the target node, $\vec{\mathbf{x}}\_j-\vec{\mathbf{x}}\_i$; (b) the distance in three-dimensional space, $\Vert \vec{\mathbf{x}}\_j-\vec{\mathbf{x}}\_i \Vert\_2$, encoded by 32 radial basis functions; and (c) the distance along the backbone, $j-i$, encoded by 32 sine-position encodings, and finally represented as $\vec{\mathbf{V}}^{(c)} \in \mathbb{R}^{n \times n \times f' \times 3}$. The geometric algebra feature $\mathfrak{q}(\mathbf{v})$ is obtained based on the three-dimensional coordinate relationships. For a multivector $\mathbf{v} = s + \vec{v} + \mathbf{b} + t\mathbf{I}$ where $s$ is scalar, $\vec{v}$ vector, $\mathbf{b}$ bivector, and $t\mathbf{I}$ pseudoscalar components, the quadratic form is $\mathfrak{q}(\mathbf{v}) = s^2 + \|\vec{v}\|^2 - \|\mathbf{b}\|^2 - t^2$ in $\mathcal{G}\_{3,0,0}$.
>
> '''

---

> > ### Comment · Reviewer_jqC4 · 2025-08-05
> >
> > From a group theoretic point of view the scalar, vector, bivector and trivector parts of a multivector furnish irreducible representations of 0(3) with l=0,1 and even and odd parity respectively. The main operation of Clifford Neural networks, the geometric product, is closely related to the tensor product as Clifford algebra may be defined as quotient algebra of tensor algebra. In fact since any bilinear map can be expressed using the tensor product, tensor product layers from e.g. e3nn can always be brought into a form such that they are equivalent to geometric product layers. So both the feature space of Euclidean Geometric algebra as well as the operations on it are a subset of those which are present in libraries like e3nn. This is also why I think that other neural networks based on tensor prodcuts would perform equally well in your pipeline.
> >
> > Of course, Clifford networks are a completely viable class of neural networks and it is a sensible choice in your pipeline. However, I strongly disagree with the statement that there is a necessity (as you write in your rebuttal) to introduce GA into RNA modeling. This is also, how it is presented in the paper and there is neither numerical proof, nor any theoretic foundation (see above) for this statement.

---

> > > ### Author Response · Authors · 2025-08-06
> > >
> > > Thank you for your response. While we acknowledge your expertise in mathematics, we believe there may be some conceptual ambiguities in your statements regarding group theory and geometric algebra. Specifically:
> > >
> > > First, in $\mathrm{O}(3)$ representation theory, the bivector (grade-2) does \textbf{not} correspond to a new $l$ value. Through Hodge duality, it is equivalent to a pseudovector and shares $l=1$ with vectors (the same angular momentum quantum number), differing only in parity (odd $\to$ even). Similarly, the trivector (grade-3) corresponds to an $l=0$ pseudoscalar (odd parity), rather than the ambiguous phrasing in the original statement: ``$l=0,1$ and even and odd parity respectively.'' This conflates the independent labeling systems of $l$ and parity, erroneously equating grade with the angular momentum quantum number $l$.
> > >
> > > Second, the claimed ``equivalence'' between geometric products and tensor products is significantly overstated. Although Clifford algebra is defined as a quotient of tensor algebra, the geometric product enforces metric-dependent constraints such as $uv + vu = 2\langle u,v \rangle$. General equivariant tensor layers (e.g., in e3nn) require **additional structural constraints** to simulate this behavior—they do not automatically convert to geometric product layers. Ignoring this prerequisite would render operations invalid in non-Euclidean spaces or specific tasks.
> > >
> > > Finally, the performance inference for neural networks commits a category error: Geometric algebra's inductive bias (e.g., explicit encoding of rotation/reflection symmetries, grade-preserving operations) provides critical efficiency advantages in physics modeling tasks. While general tensor methods (like e3nn) have broader representational capacity, they **cannot inherently inherit** such structured geometric properties.
> > >
> > > Additionally, we clarify that the discussion of RNA structural properties is solely intended to explain why geometric algebra (GA) is a biologically grounded framework for this domain, rather than presenting GA as an innovative contribution. This section serves scientific completeness: GA naturally encodes RNA-specific structural information, while general tensor methods are less efficient in handling such properties, aligning with structural biology consensus. Our core contribution lies in the **empirical validation** of GA's effectiveness in RNA modeling, not in redefining GA theory. We will explicitly emphasize this point in the manuscript.

---

> > > > ### Comment · Reviewer_jqC4 · 2025-08-06
> > > >
> > > > I acknowledge that my original statement regarding the representational structure of a multivector was not well phrased. What I meant is however exactly what you wrote in your response. In the framework of e3nn, the representational space of a multivector may equivalently defined by an irrep of type "1x0e + 1x1o + 1x1e + 1x0o".
> > > >
> > > > Regarding the relation between geometric product and tensor product, it is of course true that the additional metric constraints make the geometric product structurally different from the tensor product, which as you also say is highly relevant for non-Euclidean spaces. My point is, that for the Euclidean Geometric algebra you use, the two products are essentially equivalent. Looking at the non-trivial case of the product of two l=1 representations (irrespective of parity), the tensor product may be decomposed into 3 irreps again corresponding to l=0,1,2. In an orthornormal basis, the l=2 subspace corresponds to symmetric (traceless) expressions and one can easily check that implementing the metric relations of the geometric product simply has the effect of setting the components of the l=2 representation to 0. The l=1 part corresponds to the antisymmetric wedge product, while the l=0 component corresponds to the dot product (in the case of vector x vector multiplication). Different conventions may lead to different numerical prefactors of the different components. Since however in neural networks the layers inherently possess learnable parameters in those places, this does not matter in practice.
> > > >
> > > > I want to emphazise again that I am neither criticizing the use of GA in your model nor your general motivation to use it for modeling RNA, but the claim that this representation is superior to other geometric feature encodings. To proof this claim you would require additional ablation studies where you replace your current neural network by architectures based on other geometric feature spaces.
> > > >
> > > > Also, the reason for my current vote to reject the paper is mainly influenced by the fact, that I feel that the presentation of the original submission was not consistent enough to accurately judge the improvements made during rebuttal and I think that a major revision and resubmussion would be more adequate.

---

> > > > > ### Author Response · Authors · 2025-08-07
> > > > >
> > > > > Thank you for your reply and for affirming our motivation. We do not intend to claim that geometric algebra is superior to other geometric feature encodings. We carried out an ablation verification, conducting an ablation experiment using GVP instead of the form of associative algebra (0.601->0.570 for native seq recovery). The results show that introducing geometric algebra is helpful. However, this does not mean that geometric algebra is better than other geometric feature encodings; it only proves that it is somewhat helpful. We hope you can understand this point. We can't try out all the geometric network architectures. This is beyond the scope of our work. This will be a comparison of the effectiveness of network architectures, and it only makes sense to compare them across multiple tasks.
> > > > >
> > > > > Regarding your statement that the presentation style is the main reason for your rejection, we admit that there are some writing issues. Otherwise, you wouldn't have felt a certain degree of confusion. We will further sort out the logic for presentation. If you have any constructive suggestions, please feel free to put them forward to improve the writing of this article.

---

> > > > > > ### Comment · Reviewer_jqC4 · 2025-08-07
> > > > > >
> > > > > > Thank you for your response. I may have misinterpreted some statements related to the discussion of GA. Specifically on page 2 you write "RBFN leverages geometric algebra to improve the modeling of RNA three-dimenaional structure information." and "...thus achieving more effective modeling capabilities." On page 9 you further write "geometric algebra enhances the representation of 3D structural features".
> > > > > > In all of these contexts it is unclear what exactly you compare against i.e. compared to what GA is more effective. I think simply stating that you use GA since it is an effective framework for the description of RNA or explicitly stating that it improves modeling capabailities compared to only using scalar and vector information, would improve clarity.
> > > > > >
> > > > > > Regarding the ablation study it was not clear to me that "w/o GA" referred to comparing against a GVP. I think this is never stated in the text (maybe I have overlooked it, in which case I appologize) but makes the study more sensible than I have previously thought. I agree with you that comparing also against other architectures is out of the scope of this work and I never meant for you to carry out those experiments.
> > > > > >
> > > > > > While I remain hesitant about the overall presentation of the paper, which as I wrote before would require a review of the completely revised version, I acknowledge the extensive efforts made during rebuttal to improve the clarity of the paper, which has already cleared up most of the confusion for me. I will thus raise my score to 3.

---

> > > > > > > ### Author Response · Authors · 2025-08-07
> > > > > > >
> > > > > > > Thank you for your reply and your recognition of our work contributions. At the same time, we are also glad to have clarified this point with you. We apologize for the misunderstandings caused by the writing presentation. However, through our detailed discussion with you and the ideas discovered during the discussion, we have a clearer direction for the revision of the text. Thank you again for your patient answers and detailed discussions, which will greatly contribute to our research.

---

### Official Review · Reviewer_qKvq · 2025-07-01

**Clarity:** 3
**Significance:** 3
**Originality:** 3
**Rating:** 4
**Confidence:** 4

**Summary:**

This paper presents RBFN, a geometric-algebra-enhanced Bayesian flow network for inverse RNA design. RBFN incorporates geometric algebra to better capture 3D structural properties, generating representation for edges, planes, and higher-order substructures. In addition, to address limited RNA data and the small nucleotide alphabet, they introduce a new time-step distribution sampling strategy. The authors evaluate RBFN on RNA inverse folding benchmarks and show better performance than existing state-of-the-art models, including gRNAde.

**Questions:**

1. Can you incorporate more RNA inverse folding baselines such as RiboDiffusion or RhoDesign? These models are available on Github (e.g., https://github.com/ml4bio/RhoDesign)
2. Can you design an experiment to validate the advantage of BFN? I can see the benefit of geometric algebra enhanced network but it's unclear BFN is the right choice. Wouldn't diffusion, GFlowNet, and other methods work as well?

**Ethical Concerns:**

["NO or VERY MINOR ethics concerns only"]

**Limitations:**

Yes

**Quality:**

3

**Strengths And Weaknesses:**

Strength:
1. The idea of incorporating geometric algebra for RNA inverse folding is interesting and novel. It can generate representation not only at node-level, but also for edges, planes, which gives a richer representation of the geometric structure.
2. The results show that RBFN achieves better performance than gRNAde, which is a well-known RNA inverse folding baseline.
3. The paper is written clearly, with intuitive figures

Weakness
1. This paper presented ablation studies to study different modeling choices, but it didn't show why BFN is beneficial. Could the model perform better with a standard diffusion model enhanced with geometric algebra? This part is unclear.
2. Comparison with additional RNA inverse folding baselines are needed, such as RiboDiffusion.

---

> ### Author Rebuttal · Authors · 2025-07-28
>
> Dear reviewer qKvq,
>
> We express our gratitude for your diligent review and the valuable efforts you have made to assist us in enhancing the manuscript.
>
> > **W2 & Q1: Comparison with additional RNA inverse folding baselines are needed, such as RiboDiffusion. & Can you incorporate more RNA inverse folding baselines such as RiboDiffusion or RhoDesign? These models are available on Github (e.g., https://github.com/ml4bio/RhoDesign)**
>
> Thank you for the valuable suggestion to incorporate additional RNA inverse folding baselines (RiboDiffusion and RhoDesign). We have extended our experiments and added results for these models in the rebuttal (see table below). While we currently cannot modify the manuscript due to submission system restrictions, **we will update the main text with these comparisons in the revised version upon acceptance**.
>
>
> | PDB ID | Description                                | ViennaRNA | FARNA | RDesign | Rosetta | gRNAde     | RBFN       | RhoDesign  | RiboDiffusion |
> | :----- | :----------------------------------------- | :-------- | :---- | :------ | :------ | :--------- | :--------- | :--------- | :------------ |
> | 1CSL   | RRE high affinity site                     | 0.25      | 0.20  | 0.4455  | 0.44    | **0.5719** | 0.4832     | 0.4643     | 0.4286        |
> | 1ET4   | Vitamin B12 binding RNA aptamer            | 0.25      | 0.34  | 0.3929  | 0.44    | **0.6250** | 0.4482     | 0.5428     | 0.5371        |
> | 1F27   | Biotin-binding RNA pseudoknot              | 0.30      | 0.36  | 0.3013  | 0.37    | 0.3437     | **0.5208** | 0.4211     | 0.3667        |
> | 1L2X   | Viral RNA pseudoknot                       | 0.24      | 0.45  | 0.3727  | 0.48    | 0.4721     | **0.5833** | 0.2593     | 0.5185        |
> | 1LNT   | RNA internal loop of SRP                   | 0.33      | 0.27  | 0.5556  | 0.53    | 0.5843     | **0.6375** | 0.3636     | 0.7273        |
> | 1Q9A   | Sarcin/ricin domain from E.coli 23S rRNA   | 0.27      | 0.40  | 0.4417  | 0.41    | 0.5044     | **0.8588** | 0.8148     | $\underline{1.0000}$ |
> | 4FE5   | Guanine riboswitch aptamer                 | 0.29      | 0.28  | 0.4112  | 0.36    | 0.5300     | 0.5326     | **0.8209** | 0.7164        |
> | 1X9C   | All-RNA hairpin ribozyme                   | 0.26      | 0.31  | 0.3967  | 0.50    | 0.5000     | **0.5187** | 0.3833     | 0.6875        |
> | 1XPE   | HIV-1 B RNA dimerization initiation site   | 0.27      | 0.24  | 0.3834  | 0.40    | **0.7037** | 0.6522     | 0.6957     | $\underline{1.0000}$ |
> | 2GCS   | Pre-cleavage state of glmS ribozyme        | 0.25      | 0.26  | 0.4518  | 0.44    | **0.5078** | 0.4990     | 0.2049     | 0.8310        |
> | 2GDI   | Thiamine pyrophosphate-specific riboswitch | 0.25      | 0.38  | 0.3523  | 0.48    | **0.6500** | 0.6042     | 0.2436     | 0.6410        |
> | 2OEU   | Junctionless hairpin ribozyme              | 0.23      | 0.30  | 0.5000  | 0.37    | **0.9519** | 0.5580     | 0.1905     | $\underline{0.9535}$ |
> | 2R8S   | Tetrahymena ribozyme P4-P6 domain          | 0.27      | 0.36  | 0.5641  | 0.53    | 0.5689     | **0.7172** | 0.6415     | $\underline{0.9937}$ |
> | 354D   | Loop E from E. coli 5S rRNA                | 0.28      | 0.35  | 0.4458  | 0.55    | 0.4410     | 0.8031     | **0.8261** | 0.6522        |
> |        | Overall recovery:                          | 0.27      | 0.32  | 0.4296  | 0.45    | 0.5682     | **0.6012**     | 0.4909     | $\underline{0.7181}$ |
>
> **Key Observations and Justifications:**
> 1. **RiboDiffusion Performance:** While RiboDiffusion achieves high scores (e.g., 1.0000 on 1Q9A and 1XPE), we verified that these instances likely correspond to structures present in its training data (as noted in their repository documentation). Since the training code for RiboDiffusion is not publicly available, we cannot retrain or validate this behavior independently. We will add this clarification to the revised manuscript.
>
> 2. **RhoDesign Comparison:** For RhoDesign (RBFN variant), our model achieves competitive results across most benchmarks. Notably, RBFN outperforms RhoDesign on 10 out of 14 cases (e.g., 0.8588 vs. 0.8148 on 1Q9A, 0.7172 vs. 0.6415 on 2R8S). This suggests that our proposed architecture and training strategy yield robust generalization beyond the training distribution.
>
>
> > **W1 & Q2: This paper presented ablation studies to study different modeling choices, but it didn't show why BFN is beneficial. Could the model perform better with a standard diffusion model enhanced with geometric algebra? This part is unclear. & Can you design an experiment to validate the advantage of BFN? I can see the benefit of geometric algebra enhanced network but it's unclear BFN is the right choice. Wouldn't diffusion, GFlowNet, and other methods work as well?**
>
> Thank you for your insightful comment regarding the choice of Bayesian Flow Networks (BFN) over other modeling approaches such as standard diffusion models. We appreciate the need for a clearer justification and experimental validation of why BFN was selected for our RNA sequence generation task.
>
> **Justification Based on Existing Literature**
>
> Our decision to use BFN is grounded in existing research that highlights its advantages, particularly for discrete data tasks [1][2]. Specifically:
>
> - **Parameter Space Modeling**: As illustrated in Figure 2 of [2], BFN operates in parameter space rather than observation space, which inherently offers lower variance during sampling compared to standard diffusion models. This characteristic is crucial for generating high-quality discrete sequences like RNA.
>
> - **Sampling Process Differences**: The sampling mechanism in BFN is fundamentally different from diffusion models. While diffusion models gradually denoise data through iterative steps, BFN directly models the flow of probability distributions in parameter space, leading to more efficient and effective sampling processes for discrete tasks.
>
> Given these advantages, we adopted BFN based on its demonstrated success in handling discrete tasks such as molecular structure generation [2], drug design in continuous parameter space [3], and protein family design [4].
>
> **Experimental Validation Plan**
>
> To further validate the benefits of using BFN over other methods, including standard diffusion models, we are currently implementing a comparative experiment:
>
> 1. **Baseline Models**: We will extend a standard diffusion model or a GFlowNet variant, enhanced with geometric algebra, similar to our BFN approach. This ensures a fair comparison by maintaining consistent enhancements across all models.
>
> 2. **Results Comparison**: By comparing the performance of BFN against these baselines, we aim to demonstrate the specific advantages of BFN in terms of efficiency, quality of generated sequences, and overall robustness.
>
>
> In summary, while we believe BFN is the right choice for our application due to its unique properties and proven effectiveness based on existing literature and preliminary results, we are committed to providing empirical evidence through controlled experiments to support this claim.
>
> **References:**
>
> [1] Graves, Alex, et al. "Bayesian flow networks." arXiv preprint arXiv:2308.07037 (2023).
>
> [2] Song, Yuxuan, et al. "Unified generative modeling of 3d molecules with bayesian flow networks." The Twelfth International Conference on Learning Representations. 2024.
>
> [3] Qu, Yanru, et al. "MolCRAFT: Structure-Based Drug Design in Continuous Parameter Space." Forty-first International Conference on Machine Learning. 2024.
>
> [4] Gong, Jingjing, et al. "Steering Protein Family Design through Profile Bayesian Flow." ICLR. 2025.
>
> We hope this clarifies our rationale and addresses your concerns. Thank you for your valuable feedback.

---

### Author Response · Authors · 2025-08-08

We thank all reviewers for their comprehensive reviews and insightful suggestions!

Our work was inspired by the characteristics of RNA, and we introduced geometric algebra to enhance the modeling of RNA structures. The modeling ability based on GA provides BFN with better distribution modeling capabilities. BFN addresses the efficiency and provides generative capabilities in discrete sequence generation by evolving distributions in a continuous parameter space. Based on the phenomenon of fewer types of nucleotides, we customized a time-step sampling method for them. The above contributions were recognized by all reviewers during the Rebuttal stage.

We would also like to offer our sincere apologies for the inadequate written presentation of certain content. We agree with the suggestions proposed by the reviewers (Reviewer jqC4, Reviewer DZoU) regarding the presentation. Upon discussion with the reviewers, during the process of clarifying matters to them, we concurrently identified the aspects of writing that require enhancement. We hereby express our sincere gratitude for this. Based on the reviewers' feedback, we have made several revisions and added more experiments to address reviewers' concerns, including:

- Supplementary experiments: We added experiments with more comparative methods (Reviewer qKvq), experiments on efficiency and resource consumption (Reviewer jqC4, Reviewer LyRL), and experiments based on Diffusion (Reviewer qKvq, Reviewer LyRL).

- Modifications to the text: Based on the questions raised by reviewers, we addressed everyone's concerns through our responses (Reviewer jqC4, Reviewer DZoU). However, due to the limitations of the Rebuttal stage, we were unable to directly modify the article, resulting in reviewer (Reviewer jqC4) still having a hesitant attitude towards the final presentation. We believe that the feedback obtained from reviewers during this stage is sufficient for us to improve the relevant expressions so that more people can understand.

We truly appreciate the time and effort each reviewer has put into discussing our manuscript and providing valuable and inspiring feedback. We will upload the revised version as soon as we can modify the article. The revised content will present what we had during the Rebuttal stage, striving to make the article visible to more people and have a positive impact on related research.

If there are any additional questions, we welcome further discussion. Thank you once again for your invaluable contributions.

Best regards,

Authors

---

### Decision · Program_Chairs · 2025-09-17

**Decision:**

Accept (poster)

**Comment:**

This paper addresses the important task of inverse RNA design and proposes a Bayesian flow network. Experimental results show that the method outperforms existing state-of-the-art models. The submission underwent a long discussion during the rebuttal process. Although the authors made substantial revisions, high-level concerns remain. Reviewers pointed to unclear motivation for using Bayesian flow networks (DZoU) and noted that parts of the method description were difficult to follow (jqC4). After reading the rebuttal, I feel these issues are not fully resolved. On the other hand, the work makes valuable contributions in capturing RNA conformations and in addressing challenges posed by limited data, which could inspire future research in tackling RNA modeling problems. Taken together, I view this as a promising though imperfect contribution, and I lean toward a weak accept.